# Stochastic Subspace Descent Accelerated via Bi-fidelity Line Search

**Nuojin Cheng**  *nuojin.cheng@colorado.edu*
*Department of Applied Mathematics*
*University of Colorado Boulder*

**Alireza Doostan**  *alireza.doostan@colorado.edu*
*Smead Aerospace Engineering Sciences Department*
*University of Colorado Boulder*

**Stephen Becker**  *stephen.becker@colorado.edu*
*Department of Applied Mathematics*
*University of Colorado Boulder*

**Reviewed on OpenReview:** *https://openreview.net/forum?id=xuOQUs5YmT*

## Abstract

Efficient optimization remains a fundamental challenge across numerous scientific and engineering domains, particularly when objective function evaluations are computationally expensive and gradient information is inaccessible. While zeroth-order optimization methods address the lack of gradients, their performance often suffers due to the high cost of repeated function queries. This work introduces a bi-fidelity line search scheme tailored for zeroth-order optimization. Our method constructs a temporary surrogate model by strategically combining inexpensive low-fidelity (LF) evaluations with accurate high-fidelity (HF) evaluations of the objective function. This surrogate enables an efficient backtracking line search for step size selection, significantly reducing the number of costly HF queries required. We provide theoretical convergence guarantees for this scheme under standard assumptions. Furthermore, we integrate this bi-fidelity strategy into the stochastic subspace descent framework, proposing the bi-fidelity stochastic subspace descent (BF-SSD) algorithm. A comprehensive empirical evaluation of BF-SSD is conducted across four distinct problems: a synthetic optimization benchmark, dual-form kernel ridge regression, black-box adversarial attacks on machine learning models, and transformer-based black-box language model fine-tuning. The numerical results consistently demonstrate that BF-SSD achieves superior optimization performance, particularly in terms of solution quality obtained per HF function evaluation, when compared against relevant baseline methods. This study highlights the efficacy of integrating bi-fidelity strategies within zeroth-order optimization frameworks, positioning BF-SSD as a promising and computationally efficient approach for tackling large-scale, high-dimensional optimization problems encountered in various real-world applications.

## 1 Introduction

In this work, we are interested in the unconstrained optimization problem

$$\boldsymbol{x}^* \in \arg\min_{\boldsymbol{x} \in \mathbb{R}^D} f(\boldsymbol{x}), \tag{1.1}$$

where the objective function $f : \mathbb{R}^D \to \mathbb{R}$ is assumed to be L-smooth (i.e., its gradient $\nabla f$ is L-Lipschitz continuous). Crucially, we operate in a black-box setting where direct access to the gradient $\nabla f$ is unavailable or

computationally infeasible to obtain with low complexity (e.g., via closed-form expressions or automatic differentiation), thereby precluding the direct application of standard first-order or higher-order optimization techniques. Furthermore, we focus on high-dimensional scenarios where D is large (e.g., $D \gtrsim 100$), posing significant challenges related to computational cost and scalability for many traditional derivative-free optimization methods.

The primary focus of this work is on selecting an appropriate step size (a.k.a learning rate) $\alpha_k > 0$ for iterative descent schemes of the form:

$$\boldsymbol{x}_{k+1} = \boldsymbol{x}_k - \alpha_k \boldsymbol{v}_k,$$

where $\boldsymbol{v}_k \in \mathbb{R}^D$ represents an estimate of the true gradient $\nabla f(\boldsymbol{x}_k)$ or another suitable descent direction at iteration $k$. Selecting an appropriate step size $\alpha_k$ dynamically can significantly improve the convergence performance of the optimization process. This is illustrated in Figure 1, where an example function is optimized using different methods with and without a step size tuning scheme. However, common practices in machine learning often involve either using a fixed step size throughout the optimization or employing a predefined adaptive step size scheduling, e.g., Duchi et al. (2011). While convenient to implement, these approaches often neglect the intrinsic local geometry and characteristics of the objective function.

In contrast, classical line search methods, including exact line search and backtracking algorithms, typically yield better step sizes by incorporating information from additional objective function evaluations within each iteration. However, the cost of these additional evaluations can render such methods impractical, particularly when the computational budget is limited – a common situation in black-box optimization or when evaluating $f$ (or its gradients) is expensive.

To address this limitation, we propose a novel approach to tune the step size by leveraging multi-fidelity evaluations of the objective function. Here, multi-fidelity modeling involves utilizing two or more levels of objective function representations: the high-fidelity (HF) objective $f$, which provides accurate but expensive evaluations, and one or more low-fidelity (LF) objectives that serve as cheaper, albeit less accurate, approximations of $f$. We emphasize that this multi-fidelity setup (often bi-fidelity, involving one HF and one LF model) does not necessarily need to originate from the problem's inherent structure. Even for a problem initially defined with a single fidelity, a practitioner can potentially construct a multi-fidelity extension (e.g., by creating simplified surrogate models) to accelerate the optimization procedure, particularly through more informed step size selection.

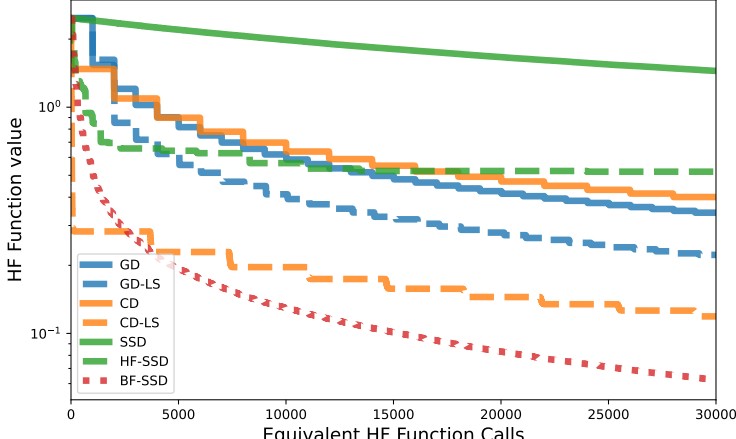

Figure 1: Gradient descent (GD), coordinate descent (CD), and stochastic subspace descent (SSD), along with their respective backtracking line search (LS) variants for step size tuning, as well as the proposed Bi-fidelity SSD (BF-SSD), are evaluated on the "worst function in the world" example, detailed in Section 4.1.

For simplicity, we focus on the bi-fidelity case, where only two fidelity levels are considered. The HF objective, $f^{\text{HF}}$, is treated as the ground-truth objective function, so we treat $f^{\text{HF}}$ and the $f$ from Equation (1.1) synonymously. We construct simple bi-fidelity surrogates *after* obtaining the gradient estimation $\boldsymbol{v}_k$. Specifically, given the LF objective $f^{\text{LF}} : \mathbb{R}^D \to \mathbb{R}$, the current position $\boldsymbol{x}_k$, $\boldsymbol{v}_k$, an initial step size $\alpha_{\max}$, and a budget $n_k$ for HF evaluations at this step, the *local* 1D surrogate of the HF objective $\varphi_k(\alpha) \coloneqq f^{\text{HF}}(\boldsymbol{x}_k - \alpha\boldsymbol{v}_k) : \mathbb{R} \to \mathbb{R}$ is constructed as

$$\tilde{\varphi}_k(\alpha; n_k) = \rho f^{\text{LF}}(\boldsymbol{x}_k - \alpha\boldsymbol{v}_k) + \tilde{\psi}_k(\alpha; n_k), \quad \alpha \in [0, \alpha_{\max}]. \tag{1.2}$$

Here, $\rho$ is a scalar, and $\tilde{\psi}_k(\cdot; n_k) : \mathbb{R} \to \mathbb{R}$ is a piecewise linear function constructed using $n_k$ HF evaluations. Once the surrogate $\tilde{\varphi}_k$ is constructed, the step size is selected using backtracking line search by (approximately) solving

$$\alpha_k = \underset{\alpha \in [0, \alpha_{\max}]}{\arg\min} \ \tilde{\varphi}_k(\alpha; n_k).$$

Assuming the scalar $\rho$ is properly chosen so that the difference

$$d(\boldsymbol{x}) \coloneqq f^{\text{HF}}(\boldsymbol{x}) - \rho f^{\text{LF}}(\boldsymbol{x})$$

is Lipschitz continuous, we show that the convergence of this descent method is guaranteed, and $K_\epsilon = \mathcal{O}(L/\epsilon)$ iterations are needed to ensure that $\min_k \|\nabla f^{\text{HF}}(\boldsymbol{x}_k)\|^2$ is $\epsilon$-small. Moreover, when the HF and LF functions are well-aligned, i.e., the Lipschitz constant $W$ of $d(\boldsymbol{x})$ is small, the required number of HF function evaluations $N_\epsilon = \mathcal{O}(WL^2/\epsilon + DL/\epsilon)$ is not large.

For implementation, we focus on high-dimensional zeroth-order optimization problems, using the stochastic subspace descent (SSD) method (Kozak et al., 2021) combined with the proposed step size tuning strategy, and call the resulting method bi-fidelity stochastic subspace descent (BF-SSD). BF-SSD demonstrates strong empirical performance across various tasks and holds great potential for future applications.

## 1.1 Related Work

**Line Search for Optimization**  Line search is a widely used method for determining step sizes in optimization algorithms. Line searches can be either exact, meaning that $\alpha$ is chosen to exactly or almost exactly minimize $f^{\text{HF}}(\boldsymbol{x}_k - \alpha\boldsymbol{v}_k)$, or inexact. Exact line searches are computationally expensive, so other than in special cases, they are rarely used in practice. Common inexact line search methods include backtracking line search (Nocedal & Wright, 1999), the Polyak step size (Polyak, 1987), spectral methods such as (Barzilai & Borwein, 1988), and learning rate scheduling (Duchi et al., 2011). Among these, backtracking line search is particularly popular due to its simplicity and explainable design, often employing stopping criteria like the Armijo and Wolfe conditions (Nocedal & Wright, 1999). However, backtracking line search increases the overall computational costs considerably due to the numerous function evaluations required at each iteration. One way to mitigate this issue is by constructing surrogate models to guide step size selection. For example, Yue & Meerbergen (2013) and Grundvig (2023) used reduced-order models to approximate the objective function during line search, while Mahsereci & Hennig (2017) and Cartis & Scheinberg (2018) employed a probabilistic Gaussian model for step size selection. Paquette & Scheinberg (2020) provided a theoretical analysis of line search in stochastic optimization. A recent work by Nguyen et al. (2025) extends the backtracking line search framework to the stochastic ISTA/FISTA method. These approaches do not account for the multi-fidelity structure of objective functions, which is the focus of this work.

**Derivative-Free and Zeroth-Order Optimization**  Derivative-free optimization refers to a family of optimization techniques that rely solely on function evaluations, without requiring gradient information, to find the optimum of an objective function. This category includes methods such as Bayesian optimization (Shahriari et al., 2015), direct search (Kolda et al., 2003), trust region methods (Conn et al., 2000), genetic algorithms (Srinivas & Patnaik, 1994), and zeroth-order optimization (Liu et al., 2020). Among these, zeroth-order methods stand out for their scalability to high-dimensional problems and reliable convergence properties. Following Liu et al. (2020), we refer to zeroth-order algorithms as the type of algorithms that approximate gradients using finite difference techniques and subsequently apply strategies similar to first-order methods. These methods have shown great promise in various machine learning applications where

objective functions are smooth but lack accessible or easy-to-compute derivatives. Recent advances include their use in solving black-box adversarial attacks (Chen et al., 2017; 2023) and fine-tuning large models, such as MeZO, S-MeZO, SubZO, etc. (Sun et al., 2022a;b; Malladi et al., 2023; Liu et al., 2024; Yu et al., 2024; Zhang et al., 2024), with minimal memory overhead. A recent work (Brilli et al., 2024) discloses the worst-case bound for derivative-free optimization with a line search-style method.

**Randomized Zeroth-Order Optimization for High-Dimensional Problems** In high-dimensional zeroth-order optimization problems, estimating gradients via finite differences can be computationally prohibitive. To address this, randomized algorithms have been proposed to reduce the cost of gradient estimation. The simultaneous perturbation stochastic approximation (SPSA) (Spall, 1992; 1998) uses Rademacher random vectors for gradient estimation, while Gaussian smoothing methods (Nesterov & Spokoiny, 2017) employ Gaussian random vectors. These algorithms typically provide gradient estimates projected onto one-dimensional subspaces. However, for certain problems, it is worth the increased number of function calls to improve gradient estimates. SSD (Kozak et al., 2021) explores this idea by projecting the gradient onto a random subspace of dimension $\ell$ for any $1 \leq \ell \leq D$, providing a more generalized framework for randomized zeroth-order optimization.

**Multi-Fidelity Modeling and Optimization** Multi-fidelity is a well-established concept in engineering and scientific computing for reducing computational costs. It has been widely applied across various domains, including aerodynamic design (Zhang et al., 2021), structural optimization (Ng & Willcox, 2014; De et al., 2020), data sampling (Cheng et al., 2024b), and uncertainty quantification (Peherstorfer et al., 2018; Cheng et al., 2024a; De & Doostan, 2022; De et al., 2023; Cheng & Doostan, 2025; Cheng, 2025). As one sub-branch, multi-fidelity optimization has been employed in hyperparameter tuning (Wu et al., 2020), accelerating Bayesian optimization (Kandasamy et al., 2016; Takeno et al., 2020), and reinforcement learning (Cutler et al., 2014) within machine learning. However, despite its relevance in settings where function evaluations are costly, its application in zeroth-order optimization remains largely unexplored (de Montbrun & Gerchinovitz, 2024) and has not been applied to any randomized zeroth-order method.

## 1.2 Contributions

In this work, we propose a multi-fidelity line-search scheme. Unlike previous approaches that utilize static surrogate (e.g., reduced-order) models, which remain fixed throughout the optimization process (Yue & Meerbergen, 2013; Grundvig, 2023), our method constructs a temporary surrogate model in each iteration, specifically *after* the gradient (or search direction) has been estimated. This allows us to focus on building a *one-dimensional surrogate* along the search direction, a significantly simpler task compared to constructing expensive $D$-dimensional surrogates for the objective function $f$. By leveraging a computationally cheaper LF model, we construct a simple, local, linear surrogate using only a small number, $n_k$, of HF evaluations per iteration. Assuming certain conditions between the LF and HF models hold, this 1D surrogate is then used to efficiently identify a suitable step size.

Specifically, this work makes the following contributions:

1. We develop the BF-SSD algorithm, a stochastic zeroth-order optimization method with a bi-fidelity line search that allows for choosing the approximation quality of the gradient by tuning $\ell$ (reducing to deterministic gradient descent when $\ell = D$).

2. When the error of the gradient estimate is negligible (e.g., $\ell$ is sufficiently large), we give specific conditions on the relation between the HF and LF functions that will guarantee convergence to a stationary point (or a global minimizer when $f$ is convex).

3. We highlight that many machine learning problems naturally have a corresponding LF model that can be used to construct a surrogate model, improving optimization efficiency. Despite its high potential, this strategy has not received sufficient attention in prior work.

4. We compare BF-SSD with other zeroth-order optimization methods on one synthetic function and the following three real-world applications:

- Kernel ridge regression with a Nyström-based LF approximation;
- Black-box image-based adversarial attacks with an LF model trained via knowledge distillation;
- Soft prompting of language models using a smaller training set to construct the bi-fidelity line search.

The rest of the paper is organized as follows. Section 2 introduces the proposed bi-fidelity line search method and provides convergence results. Section 3 details the implementation of the proposed method with SSD. Section 4 presents the experimental results, and Section 5 concludes the paper.

## 2  Line Search on Bi-fidelity Surrogate

In this section, we discuss the proposed algorithm and present the main theoretical results derived in this work. Unless specified otherwise, $\|\cdot\|$ denotes the Euclidean norm for vectors and the spectral norm (i.e., the induced 2-norm) for matrices. For simplicity and to maintain focus on our primary contributions during the theoretical analysis, we assume in the proofs that $\boldsymbol{v}_k$ provides an accurate estimate of the high-fidelity gradient, specifically $\boldsymbol{v}_k \approx \nabla f^{\mathrm{HF}}(\boldsymbol{x}_k)$. This assumption is employed solely for the theoretical development presented herein and does not hold for the practical implementation discussed in Section 3 or the numerical results presented in Section 4.

### 2.1  Algorithm

First, we define the algorithm, which consists of three steps for each iteration $k$:

1. Given the current position $\boldsymbol{x}_k \in \mathbb{R}^D$, gradient $\boldsymbol{v}_k \in \mathbb{R}^D$, and initial step size $\alpha_{\max} \in \mathbb{R}$, sample $n_k$ equi-spaced HF evaluations in $[0, \alpha_{\max}]$ and build the surrogate $\tilde{\varphi}_k : \mathbb{R} \to \mathbb{R}$ following Equation (1.2) (see Algo. 1 for details);

2. Given Armijo condition parameters $c \in (0, 1)$, $\beta \leq 1/2$, and initial step size $\alpha_{\max} \geq c/(L + cL)$, conduct bi-fidelity adjusted Armijo backtracking so that

$$
\begin{aligned}
\alpha_k &= \max_{m \in \mathbb{N}} c^m \alpha_{\max} \\
\text{s.t.} \quad &\tilde{\varphi}_k(c^m \alpha_{\max}; n_k) \leq f^{\mathrm{HF}}(\boldsymbol{x}_k) - \beta c^m \alpha_{\max} \|\boldsymbol{v}_k\|^2.
\end{aligned}
\tag{2.1}
$$

   See Algo. 2 for details.

3. Evaluate $f^{\mathrm{HF}}$ at the new point and continue the iterations.

### 2.2  Convergence Results

For convergence, we make the following assumptions:

**Assumption 2.1.** The objective function $f^{\mathrm{HF}} : \mathbb{R}^D \to \mathbb{R}$ attains its minimum $f^*$ and $\nabla f^{\mathrm{HF}}$ is $L$-Lipschitz continuous; i.e., there exists $L \in \mathbb{R}$ such that

$$
\|\nabla f^{\mathrm{HF}}(\boldsymbol{x}) - \nabla f^{\mathrm{HF}}(\boldsymbol{y})\| \leq L\|\boldsymbol{x} - \boldsymbol{y}\|, \quad \forall \boldsymbol{x}, \boldsymbol{y} \in \mathbb{R}^D.
$$

Note that Assumption 2.1 is standard for analysis of zeroth- and first-order methods. The constant $L$ must be known to the algorithm since it is used to set $\alpha_{\max}$.

**Assumption 2.2.** The difference between $f^{\mathrm{HF}}$ and $f^{\mathrm{LF}}$ is assumed to be smooth with a bounded Lipschitz constant. Specifically, we assume there exists $W, \rho \in \mathbb{R}$ such that

$$
\|\left(f^{\mathrm{HF}}(\boldsymbol{x}) - \rho f^{\mathrm{LF}}(\boldsymbol{x})\right) - \left(f^{\mathrm{HF}}(\boldsymbol{y}) - \rho f^{\mathrm{LF}}(\boldsymbol{y})\right)\| \leq W\|\boldsymbol{x} - \boldsymbol{y}\|, \quad \forall \boldsymbol{x}, \boldsymbol{y} \in \mathbb{R}^D.
$$

The Assumption 2.2 allows for $f^{\mathrm{LF}}$ to be *uncalibrated*, meaning that we do not require $f^{\mathrm{LF}}(\boldsymbol{x}) \approx f^{\mathrm{HF}}(\boldsymbol{x})$ since discrepancies can be reduced by building the surrogate $\tilde{\varphi}_k$.

Our next assumption, Assumption 2.3, is a sufficient condition that will be used in Lemma 2.7 to show that the surrogate is accurate.

**Assumption 2.3.** For each iteration $k$, we assume the number of HF evaluations, $n_k$, for building the surrogate $\tilde{\varphi}_k$ is sufficiently large such that

$$n_k \geq \frac{WL(1+c)\alpha_{\max}}{c\beta\|\boldsymbol{v}_k\|^2}, \quad \text{i.e.,} \quad n_k = \Omega\left(\frac{WL}{\|\boldsymbol{v}_k\|^2}\right),$$

where $\Omega(\cdot)$ denotes a lower bound up to a constant difference.

Using $\boldsymbol{v}_k = \nabla f(\boldsymbol{x}_k)$ and with the above assumptions satisfied and sufficiently large initial step size $\alpha_{\max} \geq c/(cL + L)$, the designed bi-fidelity line search leads to the following result:

**Theorem 2.4.** *Given an initial point $\boldsymbol{x}_0$, assuming actual gradients are accurately estimated, the algorithm in Section 2.1 generates a sequence $(\boldsymbol{x}_k)$ such that*

$$\min_{k\in\{0,\ldots,K\}} \|\nabla f^{\mathrm{HF}}(\boldsymbol{x}_k)\|^2 \leq \frac{2L(1+c)(f^{\mathrm{HF}}(\boldsymbol{x}_0) - f^*)}{(K+1)c\beta}.$$

*That is to say, $K_\epsilon = \mathcal{O}(L/\epsilon)$ iterations are required to obtain $\min_{k\leq K_\epsilon} \|\nabla f^{\mathrm{HF}}(\boldsymbol{x}_k)\|^2 \leq \epsilon$.*

**Remark 2.5.** Theorem 2.4 holds when $\boldsymbol{v}_k = \nabla f(\boldsymbol{x}_k)$. The error in approximating $\nabla f(\boldsymbol{x}_k)$ using finite difference methods with $\mathcal{O}(D)$ samples is typically negligible in comparison to the optimization error (see Kozak et al. (2023) for a precise quantitative statement for the case of SSD). Hence, assuming we accurately estimate $\boldsymbol{v}_k = \nabla f(\boldsymbol{x}_k)$ with $\mathcal{O}(D)$ samples per step, a bound for the total number of HF evaluations for $\epsilon$-convergence of the algorithm in Section 2.1 is

$$N_\epsilon = \sum_{k=1}^{K_\epsilon} (n_k + \mathcal{O}(D)) = \mathcal{O}\left(\frac{WL^2}{\epsilon^2} + \frac{DL}{\epsilon}\right). \tag{2.2}$$

**Remark 2.6.** When using *zeroth-order gradient descent*, with the same assumption that we accurately estimate $\boldsymbol{v}_k = \nabla f(\boldsymbol{x}_k)$ with $\mathcal{O}(D)$ samples per step, a bound for the total number of HF evaluations for $\epsilon$-convergence is

$$N_\epsilon = \sum_{k=1}^{K_\epsilon} (\log_{c^{-1}}(\alpha_{\max}L) + \mathcal{O}(D)) = \mathcal{O}\left(\frac{L\log(L)}{\epsilon} + \frac{DL}{\epsilon}\right). \tag{2.3}$$

The proof of Remark 2.6 follows the convergence proof of gradient descent using backtracking line search. Comparing the results in Equations (2.2) and (2.3), we observe that the advantage of using our bi-fidelity surrogate depends on the value of $W$. Notice that if $W$ is sufficiently small so that $WL \leq \epsilon \log L$, then the worst-case bound of our method is better than that of the zeroth-order gradient descent. We emphasize that our convergence result in Equation (2.2) is loose, due to the global nature of the Assumption 2.2 and difficulty in precisely describing the quality of the LF function relative to its HF counterpart. Hence, we view our convergence analysis as a reassurance that the method does converge, and rely on numerical experiments to elucidate when the method improves over baseline methods.

## 2.3 Proof of Theorem 2.4

Before the proof, we first introduce the following lemma:

**Lemma 2.7.** *With Assumption 2.2 and Assumption 2.3 satisfied, for any $\alpha \in [0, \alpha_{\max}]$, the 1D surrogate $\tilde{\varphi}_k(\alpha)$ satisfies the following bound,*

$$|\tilde{\varphi}_k(\alpha; n_k) - \varphi(\alpha)| \leq \frac{\|\boldsymbol{v}_k\|^2}{2} \min\left\{\frac{c}{(1+c)^2 L}, \frac{c\beta}{(1+c)L}, \beta\alpha_{\max}\right\} = \frac{c\beta\|\boldsymbol{v}_k\|^2}{2(1+c)L}. \tag{2.4}$$

The proof of Lemma 2.7 is in Appendix A. Following this lemma, the proof of Theorem 2.4 is given below.

*Proof.* Given Lemma 2.7, we have

$$|f^{\mathrm{HF}}(\boldsymbol{x}_{k+1}) - \tilde{\varphi}_k(\alpha_k; n_k)| = |\varphi(\alpha_k) - \tilde{\varphi}_k(\alpha_k; n_k)| \leq \frac{\|\boldsymbol{v}_k\|^2}{2} \min\left\{\frac{c}{(1+c)^2 L}, \frac{c\beta}{(1+c)L}, \beta\alpha_{\max}\right\}$$

and, using the standard descent lemma for $L$-smooth functions (guaranteed by Assumption 2.1),

$$f^{\mathrm{HF}}(\boldsymbol{x}_{k+1}) \leq f^{\mathrm{HF}}(\boldsymbol{x}_k) - \alpha_k\|\boldsymbol{v}_k\|^2 + \frac{\alpha_k^2 L}{2}\|\boldsymbol{v}_k\|^2.$$

Therefore, using the triangle inequality, the surrogate $\tilde{\varphi}_k$ is bounded as

$$\tilde{\varphi}_k(\alpha_k; n_k) \leq f^{\mathrm{HF}}(\boldsymbol{x}_{k+1}) + |f^{\mathrm{HF}}(\boldsymbol{x}_{k+1}) - \tilde{\varphi}_k(\alpha_k; n_k)|$$
$$\leq f^{\mathrm{HF}}(\boldsymbol{x}_k) + \left(-\alpha_k + \frac{\alpha_k^2 L}{2} + \frac{c}{2(1+c)^2 L}\right)\|\boldsymbol{v}_k\|^2.$$

When the step size satisfies $\alpha_k \in [c/(L+cL), 1/(L+cL)]$, the quadratic inequality $-\alpha_k + \alpha_k^2 L/2 + c/(2(1+c)^2 L) \leq -\alpha_k/2$ holds, along with the fact that $\beta \leq 1/2$, which implies the following bi-fidelity-adjusted Armijo condition

$$\tilde{\varphi}_k(\alpha_k; n_k) \leq f^{\mathrm{HF}}(\boldsymbol{x}_k) + \left(-\alpha_k + \frac{\alpha_k^2 L}{2} + \frac{c}{2(1+c)^2 L}\right)\|\boldsymbol{v}_k\|^2$$
$$\leq f^{\mathrm{HF}}(\boldsymbol{x}_k) - \frac{\alpha_k}{2}\|\boldsymbol{v}_k\|^2 \tag{2.5}$$
$$\leq f^{\mathrm{HF}}(\boldsymbol{x}_k) - \beta\alpha_k\|\boldsymbol{v}_k\|^2.$$

The last line in Equation (2.5) satisfies the bi-fidelity-adjusted Armijo condition in Equation (2.1). Therefore, the bi-fidelity backtracking either terminates immediately with $\alpha_k = \alpha_{\max}$ or else $\alpha_k \geq c/(L+cL)$, and implies

$$\tilde{\varphi}_k(\alpha_k; n_k) \leq f^{\mathrm{HF}}(\boldsymbol{x}_k) - \beta\|\boldsymbol{v}_k\|^2 \min\left\{\frac{c}{(1+c)L}, \alpha_{\max}\right\} = f^{\mathrm{HF}}(\boldsymbol{x}_k) - \frac{\beta c}{(1+c)L}\|\boldsymbol{v}_k\|^2, \tag{2.6}$$

where the last equality comes from $\alpha_{\max} \geq c/((1+c)L)$. Using $|f^{\mathrm{HF}}(\boldsymbol{x}_{k+1}) - \tilde{\varphi}_k(\alpha_k; n_k)| \leq \beta c\|\boldsymbol{v}_k\|^2/(2(1+c)L)$ (from Lemma 2.7) and Equation (2.6), we have

$$f^{\mathrm{HF}}(\boldsymbol{x}_{k+1}) \leq \tilde{\varphi}_k(\alpha_k; n_k) + \left|f^{\mathrm{HF}}(\boldsymbol{x}_{k+1}) - \tilde{\varphi}_k(\alpha_k; n_k)\right|$$
$$\leq \tilde{\varphi}_k(\alpha_k; n_k) + \frac{\beta c\|\boldsymbol{v}_k\|^2}{2(1+c)L} \tag{2.7}$$
$$\leq f^{\mathrm{HF}}(\boldsymbol{x}_k) - \frac{\beta c\|\boldsymbol{v}_k\|^2}{2(1+c)L}.$$

Equation (2.7) leads to the telescopic series

$$\frac{\beta c}{2(1+c)L}\sum_{k=0}^{K}\|\boldsymbol{v}_k\|^2 \leq \sum_{k=0}^{K}\left(f^{\mathrm{HF}}(\boldsymbol{x}_k) - f^{\mathrm{HF}}(\boldsymbol{x}_{k+1})\right)$$
$$= f^{\mathrm{HF}}(\boldsymbol{x}_0) - f^{\mathrm{HF}}(\boldsymbol{x}_{K+1}) \leq f^{\mathrm{HF}}(\boldsymbol{x}_0) - f^*.$$

Hence,

$$(K+1)\min_{k\in\{0,\dots,K\}}\|\boldsymbol{v}_k\|^2 \leq \left(\frac{\beta c}{2(1+c)L}\right)^{-1}\left(f^{\mathrm{HF}}(\boldsymbol{x}_0) - f^*\right)$$
$$= \frac{2(1+c)L}{\beta c}\left(f^{\mathrm{HF}}(\boldsymbol{x}_0) - f^*\right).$$

To guarantee $\min_{k \leq K_\epsilon} \|\nabla f^{\mathrm{HF}}(\boldsymbol{x}_k)\|^2 \leq \epsilon$, the value of $K_\epsilon$ should be

$$K_\epsilon \geq \frac{2(f^{\mathrm{HF}}(\boldsymbol{x}_0) - f^*)(1 + c)L}{\beta c \epsilon} = \mathcal{O}\left(\frac{L}{\epsilon}\right).$$

□

**Remark 2.8.** Even if $f^{\mathrm{HF}}$ is non-convex, Equation (2.7) implies that the method is a descent method, meaning $f^{\mathrm{HF}}(\boldsymbol{x}_{k+1}) \leq f^{\mathrm{HF}}(\boldsymbol{x}_k)$. Hence, after $K$ iterations, it is natural to use $\boldsymbol{x}_K$ as the output. This descent property is not enjoyed by other methods, such as subgradient descent, stochastic gradient descent, or Polyak step size gradient descent.

**Remark 2.9.** If $f^{\mathrm{HF}}$ is convex, then Theorem 2.4 implies convergence to a global minimizer. Or, if $f^{\mathrm{HF}}$ satisfies the Polyak-Lojasiewicz inequality with parameter $\mu$ (which includes some non-convex functions, as well as all strongly convex functions), then Theorem 2.4 in conjunction with the descent property implies $f^{\mathrm{HF}}(\boldsymbol{x}_K) - f^* \leq \frac{L(1+c)}{(K+1)\mu c \beta}(f^{\mathrm{HF}}(\boldsymbol{x}_0) - f^*)$, cf. Karimi et al. (2016).

### 2.4 Examples of Possible Low-Fidelity Functions

In practice, the LF function $f^{\mathrm{LF}}$ can be constructed in various ways. The most straightforward approach is when a multi-fidelity structure is intrinsically present in the problem. For example, in Cheng et al. (2024a, Section 5.1), the LF model is the exact solution to a simplified physical model that can be simulated with negligible cost, while the intended HF objective relies on relatively expensive finite element simulations. In most machine learning problems, the LF model is not explicitly given, making its construction necessary. In this section, we discuss multiple approaches for building the LF model and the associated upper bound on $W$.

**Affine Bi-Fidelity Relationship** The ideal case occurs when the HF model is an affine transformation of the LF model, i.e., $f^{\mathrm{HF}}(\boldsymbol{x}) = \rho f^{\mathrm{LF}}(\boldsymbol{x}) + c$. In this case, the Lipschitz constant $W$, as defined in Assumption 2.2, is zero, and the number of function evaluations required for convergence is proportional to the number of iterations, as $n_k = 1$ is sufficient.

**Quadratic Objective with Low-Rank LF Approximation** Consider the case where the objective is quadratic with a positive semi-definite matrix $\boldsymbol{A} \in \mathbb{R}^{D \times D}$ and denote its rank-$r$ approximation $\widetilde{\boldsymbol{A}} \in \mathbb{R}^{D \times D}$, and assume that $\mathrm{rank}(\boldsymbol{A}) \gg \mathrm{rank}(\widetilde{\boldsymbol{A}})$. The HF objective is $f^{\mathrm{HF}}(\boldsymbol{x}) = \frac{1}{2}\langle \boldsymbol{x}, \boldsymbol{A}\boldsymbol{x} \rangle + \langle \boldsymbol{x}, \boldsymbol{a} \rangle$ for $\boldsymbol{x} \in \mathcal{X}$, where $\langle \cdot, \cdot \rangle$ denotes the Euclidean inner-product, and the LF objective is $f^{\mathrm{LF}}(\boldsymbol{x}) = \frac{1}{2}\langle \boldsymbol{x}, \tilde{\boldsymbol{A}}\boldsymbol{x} \rangle + \langle \boldsymbol{x}, \boldsymbol{a} \rangle$. Assuming the input space $\mathcal{X}$ is bounded by a unit ball with radius $R$, the Lipschitz constant $W$ is upper bounded as

$$W \leq \sup_{\boldsymbol{x}} \|\nabla f^{\mathrm{HF}}(\boldsymbol{x}) - \nabla f^{\mathrm{LF}}(\boldsymbol{x})\| = \sup_{\boldsymbol{x}} \|\boldsymbol{A} - \tilde{\boldsymbol{A}}\| \cdot \|\boldsymbol{x}\| \leq \lambda_{r+1} R,$$

where $\lambda_{r+1}$ is the $(r+1)$-th largest eigenvalue of $\boldsymbol{A}$. The empirical problems in Section 4.1 and Section 4.2.1 fall into this category.

This use-case satisfies the assumptions mentioned in Section 2.2. Assumption 2.1 is satisfied since the minimum is achieved ($f^{\mathrm{HF}}$ is continuous and coercive) and the Lipschitz constant of the gradient is $L = \|\boldsymbol{A}\|$. Assumption 2.2 is satisfied using the value of $W$ above, and Assumption 2.3 can be satisfied since it is just a parameter choice.

**Full-Batch HF and Mini-Batch LF Objectives** In many machine learning settings, the objective function is expressed as a sum over a large number of terms, each corresponding to the evaluation of a loss function on an individual data sample. In this case, a natural choice for the LF objective is the summation over a smaller subset of the data. Specifically, assuming that the HF objective sums over datapoints $i = 1, \ldots, n$ and (without loss of generality, i.e., by relabeling) the LF objective sums over datapoints $i = 1, \ldots, r$ for $r \ll n$, the HF objective is $f^{\mathrm{HF}}(\boldsymbol{x}) = \frac{1}{n}\sum_{i=1}^{n} f_i(\boldsymbol{x})$, and the LF objective is $f^{\mathrm{LF}}(\boldsymbol{x}) = \frac{1}{r}\sum_{i=1}^{r} f_i(\boldsymbol{x})$. Using the triangle inequality, the Lipschitz constant $W$ is upper bounded as

$$W \leq \sup_{\boldsymbol{x}} \left\| \frac{1}{n}\sum_{i=1}^{n} \nabla f_i(\boldsymbol{x}) - \frac{1}{r}\sum_{i=1}^{r} \nabla f_i(\boldsymbol{x}) \right\| \leq \frac{n-r}{n}\left( \max_{1 \leq i \leq r} \|\nabla f_i(\boldsymbol{x})\| + \max_{1 \leq i \leq n} \|\nabla f_i(\boldsymbol{x})\| \right).$$

The terms $\|\nabla f_i(\boldsymbol{x})\|$ are bounded if each $f_i$ is Lipschitz, or equivalently, if $f_i$ is continuous and $\boldsymbol{x}$ is constrained to a compact set. An empirical problem with this setting is presented in Section 4.2.3. Our analysis is deterministic, so $W$ is a worst-case bound, but if $r$ is large and the LF subsamples are chosen uniformly at random, it would be reasonable to expect that, due to the law of large numbers, the average case behavior is significantly better than our bound.

This use-case also satisfies the assumptions mentioned in Section 2.2 under reasonable conditions. If each $\nabla f_i$ is Lipschitz continuous with constant $L_i$ (i.e., this is always true if $f_i$ is continuous and $\boldsymbol{x}$ is constrained to a compact set), then $\nabla f^{\mathrm{HF}}$ is $L$ Lipschitz continuous with $L = \frac{1}{n}\sum_{i=1}^n L_i$ via the triangle inequality, so under the mild assumption that the minimum is achieved, Assumption 2.1 is satisfied. Furthermore, Assumption 2.2 is satisfied using the value of $W$ above, and Assumption 2.3 can again be automatically satisfied since it is just a parameter choice.

**Generic Case**   Finally, we consider the most general case, without assuming specific relationships between the HF and LF objectives. By assuming the Lipschitz continuity of both the HF and LF objectives, $W$ can be bounded as

$$W = \|f^{\mathrm{HF}}(\boldsymbol{x}) - \rho f^{\mathrm{LF}}(\boldsymbol{x})\|_L \le \|f^{\mathrm{HF}}(\boldsymbol{x})\|_L + |\rho| \cdot \|f^{\mathrm{LF}}(\boldsymbol{x})\|_L,$$

for any choice of $\rho$, where $\|\cdot\|_L$ denotes the Lipschitz constant. The proportionality $\rho$ should not be chosen to minimize this bound (since that leads to $\rho = 0$) but can instead be chosen by any heuristic, such as the one used in control variate techniques (Gorodetsky et al., 2020) where $\rho = -\hat{c}/\hat{v}$ where $\hat{c}$ is an estimate of the covariance between $f^{\mathrm{HF}}$ and $f^{\mathrm{LF}}$, and $\hat{v}$ is an estimate of the variance of $f^{\mathrm{HF}}$.

# 3   Bi-Fidelity Line Search with Stochastic Subspace Descent

In this manuscript, we focus on zeroth-order optimization, utilizing stochastic subspace descent (SSD) as the implementation method. Following the algorithmic steps introduced in Section 2.1, combined with SSD, the entire process is divided into three main (iteratively implemented) components: gradient estimation to construct $\boldsymbol{v}_k$, bi-fidelity surrogate construction, and Armijo backtracking on the surrogate.

**Gradient Estimation**   SSD employs a random projection matrix $\boldsymbol{P}_k \in \mathbb{R}^{D \times \ell}$ with $\ell \ll D$. The random matrix $\boldsymbol{P}_k$ satisfies the properties $\mathbb{E}[\boldsymbol{P}_k \boldsymbol{P}_k^\top] = \boldsymbol{I}_D$ and $\boldsymbol{P}_k^\top \boldsymbol{P}_k = (D/\ell)\boldsymbol{I}_\ell$. A common choice for $\boldsymbol{P}_k$ is based on the Haar measure, where $\boldsymbol{P}_k$ is derived from the Gram-Schmidt orthogonalization of a random Gaussian matrix. The gradient estimation is given by $\boldsymbol{v}_k = \boldsymbol{P}_k \boldsymbol{g}_k$, where $\boldsymbol{g}_k$ is the finite difference estimator of the gradient:

$$\boldsymbol{g}_k := \left[ \frac{f^{\mathrm{HF}}(\boldsymbol{x}_k + \Delta \boldsymbol{p}_1) - f^{\mathrm{HF}}(\boldsymbol{x}_k)}{\Delta}, \frac{f^{\mathrm{HF}}(\boldsymbol{x}_k + \Delta \boldsymbol{p}_2) - f^{\mathrm{HF}}(\boldsymbol{x}_k)}{\Delta}, \ldots, \frac{f^{\mathrm{HF}}(\boldsymbol{x}_k + \Delta \boldsymbol{p}_\ell) - f^{\mathrm{HF}}(\boldsymbol{x}_k)}{\Delta} \right]^\top, \quad (3.1)$$

where $\Delta \in \mathbb{R}$ is a small step size and $\boldsymbol{p}_i$ is the $i$-th column of $\boldsymbol{P}_k$. Estimating $\boldsymbol{v}_k$ using Equation (3.1) requires $\ell$ function evaluations – a more accurate $\mathcal{O}(\Delta^2)$ approximation is also possible at the cost of $2\ell$ function evaluations if more than 8 digits of precision are needed. Up to the finite-difference error, $\boldsymbol{g}_k \approx \boldsymbol{P}_k^\top \nabla f^{\mathrm{HF}}(\boldsymbol{x}_k)$ so that $\boldsymbol{v}_k \approx \boldsymbol{P}_k \boldsymbol{P}_k^\top \nabla f^{\mathrm{HF}}(\boldsymbol{x}_k)$, hence $\mathbb{E}[\boldsymbol{v}_k] \approx \nabla f^{\mathrm{HF}}(\boldsymbol{x}_k)$. It is also possible to construct the same estimator without reference to the Haar measure by rewriting $\boldsymbol{v}_k$ as $\boldsymbol{v}_k = \mathrm{proj}_{\mathrm{col}(\boldsymbol{Q}_k)}(\nabla f^{\mathrm{HF}}(\boldsymbol{x}_k))$ where $\boldsymbol{Q}_k \in \mathbb{R}^{D \times \ell}$ is any random matrix with independent columns from an isotropic probability distribution (such as the standard normal).

**Surrogate Construction**   Given the estimated gradient $\boldsymbol{v}_k$ and the current position $\boldsymbol{x}_k$, the goal of surrogate construction is to build $\tilde{\varphi}_k$, denoted as

$$\tilde{\varphi}_k(\alpha) := \rho f^{\mathrm{LF}}(\boldsymbol{x}_k + \alpha \boldsymbol{v}_k) + \tilde{\psi}_k(\alpha). \tag{3.2}$$

The analysis in Section 2 assumes that $\rho$ is known and fixed. However, in practice, $\rho$ is tuned for better performance. In our application, we set $\rho_k = f^{\mathrm{HF}}(\boldsymbol{x}_k)/f^{\mathrm{LF}}(\boldsymbol{x}_k)$. We model $\tilde{\psi}_k$ as a piecewise linear function using $n_k$ additional HF evaluations at equispaced points $\{0, \tilde{\alpha}_1, \ldots, \tilde{\alpha}_{n_k} = \alpha_{\max}\}$. Specifically,

$$\tilde{\psi}_k(\alpha) = \frac{h - \alpha}{h} \psi(\tilde{\alpha}_{j-1}) + \frac{\alpha}{h} \psi(\tilde{\alpha}_j), \quad \alpha \in [\tilde{\alpha}_{j-1}, \tilde{\alpha}_j], \quad j = 1, \ldots, n_k, \tag{3.3}$$

where $h = \alpha_{\max}/n_k$ and $\psi(\alpha) = \phi(\alpha) - f^{\mathrm{LF}}(\boldsymbol{x}_k - \alpha\boldsymbol{v}_k)$. This piecewise linear interpolation is a simple, yet effective, approach for interpolating $\varphi$ in 1D and satisfies the bounds in Lemma 2.7 given sufficient $n_k$. The detailed algorithm is presented in Algorithm 1.

Figure 2 illustrates the bi-fidelity backtracking line search process using the example problem in Section 4.2.1. The blue curve represents the bi-fidelity surrogate model $\tilde{\varphi}_k$ approximating the HF function $\varphi$ (red curve). Rather than performing the line search directly on the computationally expensive HF function (red dots), the method utilizes the surrogate $\tilde{\varphi}_k$ to estimate an optimal step size. While this surrogate is an approximation and may require more surrogate function evaluations during the search itself, it substantially reduces the computational cost of line search. In this example, the cost is decreased from 4 HF function calls (for a direct search) to only 1 HF call (to build the surrogate) combined with 6 LF function calls.

---

**Algorithm 1:** Surrogate Construction

---

**Input:** $f^{\mathrm{LF}}, f^{\mathrm{HF}}, \boldsymbol{x}_k, \boldsymbol{v}_k, n_k \in \mathbb{N}, \alpha_{\max} > 0$
**Output:** 1D surrogate $\tilde{\varphi}_k$
1: Define $\{(\tilde{\alpha}_j, \varphi(\tilde{\alpha}_j))\}_{j=0}^{n_k}$ as equispaced points between 0 and $\alpha_{\max}$ (including endpoints), and compute HF evaluations $\varphi(\tilde{\alpha}_j) \leftarrow f^{\mathrm{HF}}(\boldsymbol{x}_k + \tilde{\alpha}_j\boldsymbol{v}_k)$;
2: $\rho_k \leftarrow f^{\mathrm{HF}}(\boldsymbol{x}_k)/f^{\mathrm{LF}}(\boldsymbol{x}_k)$;
3: $\psi(\tilde{\alpha}_j) \leftarrow \varphi(\tilde{\alpha}_j) - \rho_k f^{\mathrm{LF}}(\boldsymbol{x}_k + \tilde{\alpha}_j\boldsymbol{v}_k), \quad j = 1, \ldots, n_k$;
4: Construct piecewise linear function $\tilde{\psi}_k$ using Equation (3.3);
5: Return $\tilde{\varphi}_k$ using Equation (3.2).

---

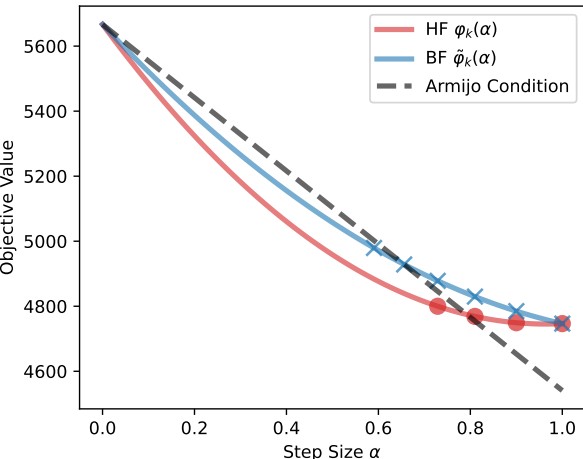

Figure 2: Illustration of the bi-fidelity backtracking line search process using the example problem in Section 4.2.1. The blue curve represents the bi-fidelity surrogate model $\tilde{\varphi}_k$ approximating the HF function $\varphi$ (red curve). It significantly lowers computational cost (e.g., reducing 4 HF calls to 1 HF + 6 LF calls).

**Armijo Backtracking on the Surrogate**  Based on the criteria in Equation (2.1), we set the maximum number of iterations for testing the Armijo condition to $M \in \mathbb{N}$. The detailed procedure is presented in Algorithm 2.

**Convergence Analysis of SSD with Line Search**  The convergence results of SSD with line search (on the exact $\varphi(\alpha)$) are presented in Appendix B, under three separate scenarios: strongly convex, convex, and non-convex. The proof shows that, in the SSD with line search setting, the value of $\beta$ can be set as $\ell/2D$.

---

**Algorithm 2:** BF-Backtracking

**Input:** $\tilde{\varphi}_k, \beta > 0, c \in (0,1), \alpha_{\max} > 0, \boldsymbol{v}_k, M \in \mathbb{N}$        `// typical value of ` $c \approx 0.9$

**Output:** Step size $\alpha_k$

 1: Initialize $\alpha_k \leftarrow \alpha_{\max}$;
 2: **for** $m = 0 : M$ **do**
 3:     **if** $\tilde{\varphi}_k(\alpha_k) \leq f^{\mathrm{HF}}(\boldsymbol{x}_k) - \alpha_k \beta \|\tilde{\boldsymbol{v}}_k\|^2$ **then**
 4:        Break;
 5:     **else**
 6:        $\alpha_k \leftarrow c\alpha_k$;
 7:     **end if**
 8: **end for**
 9: Return $\alpha_k$;

---

The proposed bi-fidelity line search algorithm, combined with SSD, will be referred to as bi-fidelity SSD (BF-SSD), and is summarized in Algorithm 3. Our theory covers either $\ell = D$ with bi-fidelity line search (Thm. 2.4) or $1 \leq \ell \leq D$ with HF line search (Appendix B) under strongly convex, convex or non-convex settings. Combining the two analyses is fairly complicated, and we defer it to a future study. For reference, the bi-fidelity result is in Theorem 2.4 and below is one of the results from Appendix B.3 for the $1 \leq \ell \leq D$ with HF line search case, in particular, the result covering the non-convex case:

**Theorem 3.1** (proof in Appendix B.3)**.** *With Assumption B.5 holding and backtracking implemented for line search, we have*

$$\min_{k \in \{0,\dots,K\}} \mathbb{E}[\|\nabla f(\boldsymbol{x}_k)\|^2] \leq \max\left\{ \frac{(f(\boldsymbol{x}_0) - f^*)}{(K+1)\beta\alpha_{\max}}, \frac{DL(f(\boldsymbol{x}_0) - f^*)}{(K+1)\ell c\beta} \right\}.$$

*That is, $k = \mathcal{O}(1/(\epsilon\beta\alpha_{\max}) + DL/(\epsilon\ell c\beta))$ iterations are required to achieve $\mathbb{E}\|\nabla f(\boldsymbol{x}_k)\|^2 \leq \epsilon$.*

For practical purposes, since the parameters in Assumption 2.3 are often unknown, we set $\rho_k$ as described above and choose $n_k = 1$ to minimize the cost of generating $\tilde{\varphi}_k(\alpha)$. This choice also demonstrates excellent empirical performance across all HF and LF pairs we have examined. We leave it as future work for adaptively choosing $n_k$, but suggest one possible scheme inspired by similar methods used in trust-region algorithms: starting with a small $n_k$, we find the stepsize via line search on the surrogate, as usual, and then compare the actual function value of the high-fidelity function with the predicted value from the surrogate. If the predicted decrease from the surrogate was overly optimistic, we then rebuild the surrogate with a larger $n_k$ (re-using existing samples) and repeat.

---

**Algorithm 3:** Bi-Fidelity Line Search SSD Algorithm

**Input:** $f^{\mathrm{HF}}, f^{\mathrm{LF}}, \ell, c, M, \alpha_{\max}, n$        `// by default, ` $n = 1$ ` and ` $\beta = \ell/2D$

**Output:** HF minimum value

 1: Initialize $\boldsymbol{x}_0$ and set of HF values $\mathcal{D} = \{f^{\mathrm{HF}}(\boldsymbol{x}_0)\}$
 2: $\beta \leftarrow \ell/2d$;
 3: **for** $k = 0 : K$ **do**
 4:     Sample random matrix $\boldsymbol{P}_k$;
 5:     Approximate $\tilde{\boldsymbol{v}}_k \approx \boldsymbol{P}_k \boldsymbol{P}_k^T \nabla f(\boldsymbol{x}_k)$ using finite difference ($\ell$ HF evaluations);
 6:     Normalize $\boldsymbol{v}_k \leftarrow \tilde{\boldsymbol{v}}_k / \|\tilde{\boldsymbol{v}}_k\|$;
 7:     Construct $\tilde{\varphi}_k \leftarrow$ surrogate-construction$(f^{\mathrm{LF}}, f^{\mathrm{HF}}, \boldsymbol{x}_k, \boldsymbol{v}_k, n, \alpha_{\max})$ ($n$ HF evaluations);
 8:     $\alpha_k \leftarrow$ BF-backtracking$(\tilde{\varphi}_k, \beta, c, \alpha_{\max}, \boldsymbol{v}_k, M)$;
 9:     Update $\boldsymbol{x}_{k+1} \leftarrow \boldsymbol{x}_k - \alpha_k \boldsymbol{v}_k$;
10:     Evaluate $f^{\mathrm{HF}}(\boldsymbol{x}_{k+1})$ and update $\mathcal{D}$;
11: **end for**
12: Return $\min \mathcal{D}$;

---

## 4 Empirical Experiments

In this section, we evaluate the proposed BF-SSD Algorithm 3 on four distinct: one synthetic optimization problem discussed in Section 4.1 and three machine learning-related problems across diverse scenarios presented in Section 4.2. These include dual-form kernel ridge regression (Section 4.2.1), black-box adversarial attacks (Section 4.2.2), and transformer-based black-box language model fine-tuning (soft prompting) in Section 4.2.3. We demonstrate that the BF-SSD algorithm consistently outperforms competing methods. To illustrate these advantages, we compare BF-SSD against the following baseline algorithms:

- **Gradient descent (GD)**: A zeroth-order gradient descent method, where the full-batch gradient is estimated using forward differences, and a fixed step size is used.

- **Nesterov accelerated gradient descent (NAG)**: An accelerated zeroth-order gradient descent method by Nesterov that incorporates a decaying momentum term, which often leads to faster convergence. The gradient is estimated using forward differences.

- **Coordinate descent (CD)**: Iteratively and individually optimizes each coordinate using finite-difference estimated coordinate gradients.

- **Stochastic subspace descent with fixed step size (FS-SSD)**: The standard stochastic subspace descent method, which samples subspaces from the Haar measure and uses a fixed step size.

- **Simultaneous perturbation stochastic approximation (SPSA)**: A randomized optimization method using a Hadamard random variable to estimate the gradient, as proposed by Spall (1992) and with step sizes as described in Spall (1998); this is a time-tested, well-established zeroth-order method.

- **Gaussian smoothing (GS)**: A method popularized by Nesterov & Spokoiny (2017), which is nearly equivalent to SSD with $\ell = 1$, and uses a fixed step size.

- **High-Fidelity stochastic subspace descent (HF-SSD)**: A single-fidelity SSD method that utilizes a high-fidelity function for backtracking line search, with its convergence analysis detailed in Appendix B.

- **Variance-reduced stochastic subspace descent (VR-SSD)**: A variance-reduced version of the SSD method inspired by SVRG (Johnson & Zhang, 2013), as described in the technical report Kozak et al. (2019, Section 2.2) of the SSD authors.

- **Bi-fidelity stochastic subspace descent (BF-SSD)**: The proposed method detailed in Section 3.

The performance of the optimizers is assessed based on the number of HF objective function evaluations required, accounting for LF calls (in terms of fractional equivalent HF function calls) as appropriate. While we also measured wall-clock time, the results strongly aligned with the equivalent HF call data. Therefore, for clarity, we present performance primarily in terms of equivalent HF calls.

### 4.1 Synthetic Problem: Worst Function in the World

In this section, we investigate the performance of our proposed BF-SSD algorithm on the "worst function in the world" (Nesterov, 2013). With a fixed Lipschitz constant $L > 0$, the function is

$$f(\boldsymbol{x}; r, L) = L \left( \frac{x_1^2 + \sum_{i=1}^{r-1}(x_i - x_{i+1})^2 + x_r^2}{8} - \frac{x_1}{4} \right) - \frac{Lr}{8(r+1)},$$

where $x_i$ denotes the $i^{\text{th}}$ entry of the input $\boldsymbol{x}$ and $r < D$ is a constant integer defining the intrinsic dimension of the problem. The function is convex with the global minimum value 0. The Lipschitz constant of the gradient of this function is $L$. Nesterov (2013) has shown that a wide range of iterative first-order methods perform poorly when minimizing $f(\boldsymbol{x}; r, L)$ with initial point $\boldsymbol{x}_0 = \boldsymbol{0}$.

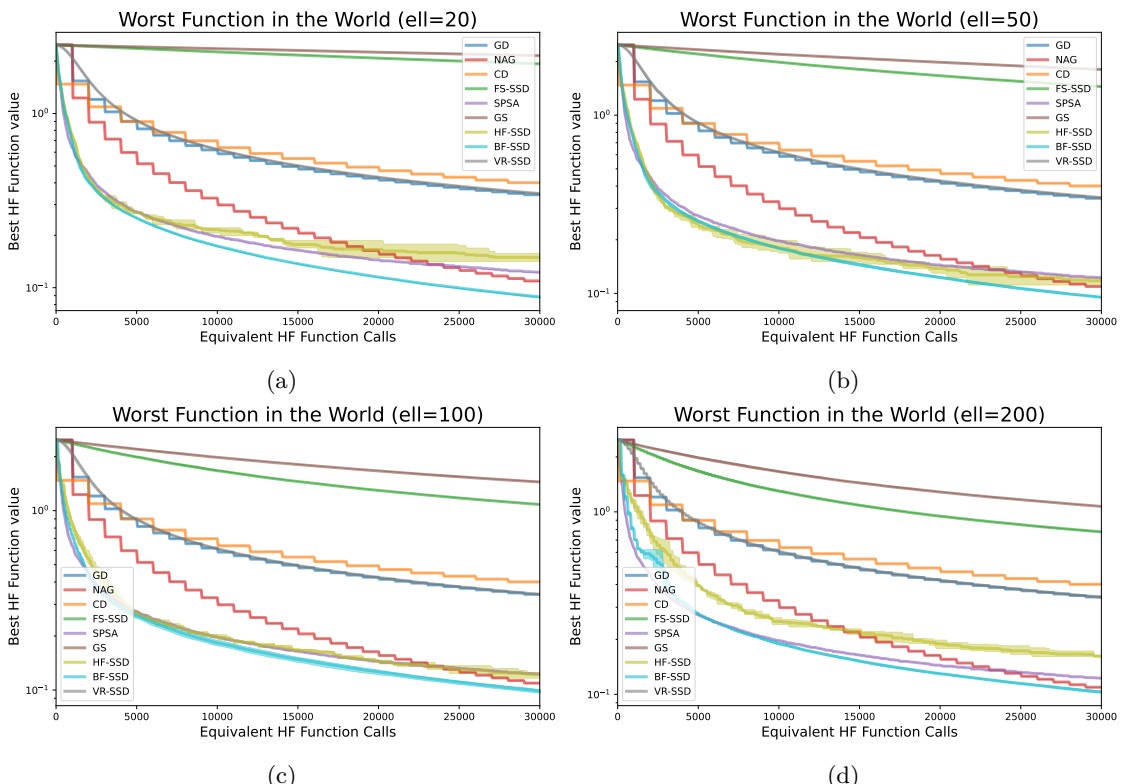

Figure 3: Convergence performance for different optimizers. The $x$-axis is the equivalent number of HF function evaluations, including the number of LF function evaluations based on the cost ratio $r_L/r_H$. The $y$-axis is the HF function evaluation value at the current stage. We investigate the results when $\ell = 20, 50, 100, 200$ with $r_L = 2$, $r_H = 100$. The corresponding results are presented with their titles indicating the specific choices. The shadow regions are the area between the best and the worst behavior over 10 trials.

We set the dimension $D = 1,000$, $\ell = 20$, and $L = 20$. The intrinsic dimensions of the LF and HF functions are $r_H$ and $r_L$, respectively. We choose $r_L \ll r_H$ and assume the computational cost ratio between HF and LF evaluations is $r_H : r_L$. For Gradient Descent, we choose the standard step size of $1/L = 0.05$, and for the GS and SSD-based methods, the step size is $\ell/(LD)$. The backtracking parameter is $\beta = \ell/(2D)$. The hyperparameter study is conducted according to different values of $c \in \{0.8, 0.9, 0.99\}$ and $\ell \in \{5, 10, 20\}$. All the experiments are repeated 10 times, with shaded regions denoting the worst and the best performance over 10 trials.

Figure 3 illustrates the performance of various optimizers across different values of $\ell$. Detailed results for $\ell = 20$ and $c = 0.99$ at $N$ from 500 to 8,000 are presented in Table 1, while additional comparisons across different $\ell$ and $c$ configurations are included in Table 4. These results show that BF-SSD consistently outperforms the other optimizers in most scenarios. For different SSD methods, the effect of $\ell$ on the final performance varies. Large values of $\ell$ improve the optimization results for FS-SSD and VR-SSD, while HF-SSD and BF-SSD prefer relatively smaller $\ell$, as highlighted in Table 2.

| Method | Equivalent HF function evaluations $N$ | | | | |
|---|---|---|---|---|---|
| | $N = 100$ | $N = 1000$ | $N = 10000$ | $N = 20000$ | $N = 30000$ |
| GD | $2.48 \pm 0.00$ | $2.48 \pm 0.00$ | $0.62 \pm 0.00$ | $0.43 \pm 0.00$ | $0.34 \pm 0.00$ |
| CD | $\mathbf{1.48 \pm 0.00}$ | $1.48 \pm 0.00$ | $0.70 \pm 0.00$ | $0.49 \pm 0.00$ | $0.40 \pm 0.00$ |
| NAG | $2.48 \pm 0.00$ | $2.48 \pm 0.00$ | $0.33 \pm 0.00$ | $0.16 \pm 0.00$ | $0.11 \pm 0.00$ |
| FS-SSD | $2.47 \pm 0.00$ | $2.45 \pm 0.00$ | $2.26 \pm 0.00$ | $2.08 \pm 0.00$ | $1.93 \pm 0.00$ |
| SPSA | $1.93 \pm 0.00$ | $\mathbf{0.66 \pm 0.00}$ | $0.20 \pm 0.00$ | $0.15 \pm 0.00$ | $0.12 \pm 0.00$ |
| GS | $2.47 \pm 0.00$ | $2.46 \pm 0.00$ | $2.36 \pm 0.00$ | $2.25 \pm 0.00$ | $2.15 \pm 0.00$ |
| HF-SSD | $1.99 \pm 0.05$ | $0.75 \pm 0.01$ | $0.40 \pm 0.21$ | $0.37 \pm 0.21$ | $0.20 \pm 0.06$ |
| BF-SSD | $2.00 \pm 0.08$ | $0.68 \pm 0.02$ | $\mathbf{0.17 \pm 0.00}$ | $\mathbf{0.11 \pm 0.00}$ | $\mathbf{0.09 \pm 0.00}$ |
| VR-SSD | $2.47 \pm 0.00$ | $2.09 \pm 0.01$ | $0.62 \pm 0.00$ | $0.43 \pm 0.00$ | $0.35 \pm 0.00$ |

Table 1: Performance values (mean $\pm$ std over 10 runs) showing the objective function for different optimization methods at various HF function evaluations $N$ with $\ell = 20$ and $c = 0.9$. The minimum values in each *column* are highlighted in bold.

Table 2: Comparison of SSD methods for different values of $\ell$ (Mean $\pm$ Std at $N = 20,000$). Bold values indicate the minimum mean for each SSD method, i.e., across each row.

| Method | $\ell = 20$ | $\ell = 50$ | $\ell = 100$ | $\ell = 200$ |
|---|---|---|---|---|
| FS-SSD | $2.0787 \pm 0.0013$ | $1.6621 \pm 0.0003$ | $1.2942 \pm 0.0022$ | $\mathbf{0.9473 \pm 0.0027}$ |
| HF-SSD | $0.3696 \pm 0.2109$ | $\mathbf{0.1482 \pm 0.0098}$ | $0.1574 \pm 0.0157$ | $0.3696 \pm 0.2368$ |
| BF-SSD | $\mathbf{0.1143 \pm 0.0005}$ | $0.1226 \pm 0.0015$ | $0.1260 \pm 0.0032$ | $0.1298 \pm 0.0013$ |
| VR-SSD | $0.4309 \pm 0.0025$ | $0.4281 \pm 0.0013$ | $0.4238 \pm 0.0009$ | $\mathbf{0.4226 \pm 0.0013}$ |

## 4.2 Zeroth-Order Optimization for Machine Learning Problems

Next, we contrast the BF-SSD optimization results against other competing zeroth-order methods. Besides showing the advantages of the BF-SSD, we also show that it is often convenient to design a cheap LF model in many machine learning problems that can be leveraged to accelerate the convergence.

### 4.2.1 Dual Form of Kernel Ridge Regression

Consider a kernel ridge regression problem as follows. By the representer theorem, given data $\{(\boldsymbol{x}_i, y_i)\}_{i=1}^{D}$ and a kernel function $\kappa : \mathbb{R}^{\tilde{m}} \times \mathbb{R}^{\tilde{m}} \to \mathbb{R}$, the goal is to find the coefficients $\tilde{\boldsymbol{\alpha}}$ such that

$$f_{\text{predict}}(\boldsymbol{x}) = \sum_{i=1}^{D} \tilde{\alpha}_i k(\boldsymbol{x}, \boldsymbol{x}_i). \tag{4.1}$$

One way to compute the coefficients is to solve the dual form of the kernel ridge regression,

$$\tilde{\boldsymbol{\alpha}}^* = \arg\min_{\boldsymbol{\alpha}} \boldsymbol{\alpha}^T \boldsymbol{K} \boldsymbol{\alpha} - 2\langle \boldsymbol{\alpha}, \boldsymbol{y} \rangle + \lambda \|\boldsymbol{\alpha}\|^2, \tag{4.2}$$

where $\boldsymbol{K}$ is the kernel matrix with $[\boldsymbol{K}]_{i,j} = \kappa(\boldsymbol{x}_i, \boldsymbol{x}_j)$, $[\boldsymbol{y}]_i = y_i$, and $\lambda$ is a positive scalar regularization parameter. The solution of Equation (4.2) can be explicitly represented as

$$\tilde{\boldsymbol{\alpha}}^* = (\boldsymbol{K} + \lambda \boldsymbol{I})^{-1} \boldsymbol{y}.$$

However, solving the explicit solution involves inverting the matrix $\boldsymbol{K} + \lambda \boldsymbol{I}$, which takes $\mathcal{O}(D^3)$ and can be extremely expensive when $D$ is large. When $D$ is sufficiently large, evaluating the function in Equation (4.1)

takes $\mathcal{O}(D^2)$ and becomes expensive. Therefore, an alternative approach to solve this problem is to build a low-rank approximation (surrogate) of the kernel matrix $\boldsymbol{K}$, which we adopt for evaluating the LF objective. To do this, we employ the Nyströem method, which finds a subset $\mathcal{S} \subset [1, \ldots, D]$ with size $l \ll D$ and builds the kernel surrogate $\tilde{\boldsymbol{K}} = \boldsymbol{K}[:, \mathcal{S}](\boldsymbol{K}[\mathcal{S}, \mathcal{S}])^{-1}\boldsymbol{K}[\mathcal{S}, :]$. By implementing the Nyströem method, the complexity of evaluating the objective function is reduced to $\mathcal{O}(lD)$. Therefore, the ratio of computational cost between HF and LF function evaluation is $D/l$.

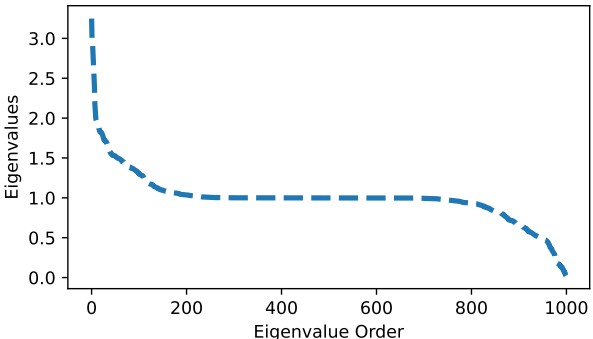

Figure 4: The eigenvalues of the kernel matrix implemented in Equation (4.2).

We pretend the problem in a black-box format, where access to the HF function is only available through an API, thus hiding $\boldsymbol{y}$ (and/or $\boldsymbol{K}$) in Equation (4.2) and making derivative information inaccessible. In this case, we assume the values of $\boldsymbol{y}$ are unavailable for privacy reasons. For the regression data, we select the first $D = 1,000$ samples from the California housing dataset provided in the scikit-learn library (Pedregosa et al., 2011). A subset $\mathcal{S}$ with size $l = 10$ is randomly selected for the Nyströem approximation. We use a Gaussian (RBF) kernel with lengthscale 1.0 to generate the corresponding kernel matrix $\boldsymbol{K}$. Figure 4 shows the decay of eigenvalues for $\boldsymbol{K}$, with a rapid drop, especially within the first 100 eigenvalues, due to the Gaussian kernel's properties. This fast decay motivates our focus on cases where the values of $\ell$ are below 100. The starting point $\boldsymbol{x}_0$ is set at the origin $\boldsymbol{0}$.

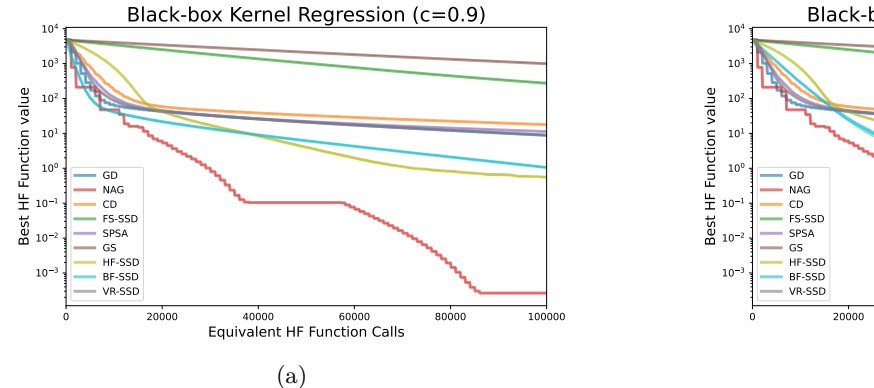
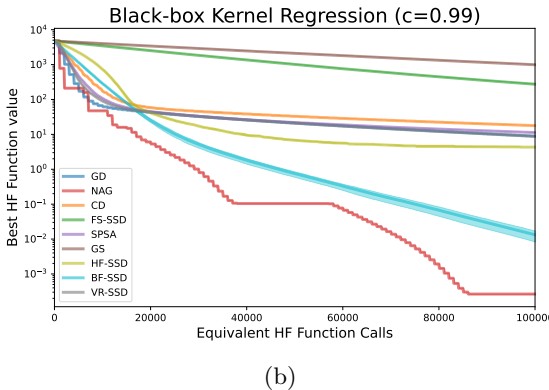

(a)                                                                 (b)

Figure 5: Similar to Figure 3, we compare the optimizer performances with varying parameters $\ell = 100$ and $c \in \{0.9, 0.99\}$. The corresponding results are presented with their titles indicating the specific choices. The shadow regions are the area between the best and the worst behavior over 10 trials.

The results of kernel ridge regression are shown in Figure 5, with values of $c$ chosen from 0.9 and 0.99, and the value of $\ell$ is fixed as 100. According to these results, BF-SSD shows advantages over other methods except NAG and HF-SSD in Figure 5a. When the backtracking factor $c$ decreases, the step sizes determined by the

backtracking method become more conservative, leading to suboptimal results, especially for BF-SSD. We also implement different combinations of $c$ and $\ell$ and collect the SSD performances in Table 3. The results suggest that BF-SSD outperforms other SSD methods for most cases, and larger values of $\ell$ and $c$ improve the performance of BF-SSD.

| $c$ | $\ell$ | FS-SSD | HF-SSD | VR-SSD | BF-SSD |
|---|---|---|---|---|---|
| | 10 | $3500.91 \pm 5.74$ | $9.49 \pm 0.21$ | $23.31 \pm 0.83$ | $\mathbf{8.18 \pm 0.60}$ |
| 0.9 | 50 | $1015.82 \pm 5.90$ | $\mathbf{4.01 \pm 0.13}$ | $20.87 \pm 0.55$ | $6.13 \pm 0.22$ |
| | 100 | $270.95 \pm 7.07$ | $\mathbf{5.49 \pm 0.18}$ | $21.56 \pm 0.69$ | $6.22 \pm 0.25$ |
| | 10 | $3493.88 \pm 11.27$ | $18.39 \pm 0.28$ | $23.20 \pm 0.30$ | $\mathbf{2.94 \pm 0.49}$ |
| 0.95 | 50 | $1013.22 \pm 8.76$ | $5.33 \pm 0.35$ | $21.44 \pm 1.37$ | $\mathbf{2.22 \pm 0.21}$ |
| | 100 | $270.36 \pm 2.62$ | $6.15 \pm 0.17$ | $21.02 \pm 0.98$ | $\mathbf{2.28 \pm 0.10}$ |
| | 10 | $3503.99 \pm 3.77$ | $29.55 \pm 0.05$ | $23.35 \pm 0.38$ | $\mathbf{1.40 \pm 0.29}$ |
| 0.99 | 50 | $1010.92 \pm 7.17$ | $6.95 \pm 0.19$ | $20.90 \pm 0.31$ | $\mathbf{0.75 \pm 0.12}$ |
| | 100 | $273.64 \pm 3.41$ | $6.26 \pm 0.08$ | $21.38 \pm 0.46$ | $\mathbf{0.79 \pm 0.24}$ |

Table 3: Black-box kernel ridge regression HF function values (mean $\pm$ std) for FS-SSD, HF-SSD, VR-SSD, and BF-SSD at various combinations of $\ell$ and $c$ at $N = 50,000$. Considering uncertainties, the minimum values in each row are highlighted in bold.

### 4.2.2 Black-box Adversarial Attack on MNIST

In practice, especially in explainable AI (XAI), researchers have found that many deep learning models are not robust to data noise. Specifically, if test data is contaminated by a small perturbation imperceptible to humans, many previously well-performing deep learning models fail to produce reasonable results (Goodfellow et al., 2014). Generating such biased noise to confuse a trained neural network model is usually referred to as "attack" in adversarially robust training. This need not be a "black hat" activity, as it can be used as part of hardening a system to prevent these attacks in the future.

There are primarily two types of attacks: one is a white-box attack, in which we have knowledge of the model and inject the adversarial noise to confuse the given model. The standard approach under this scenario is to generate the shift in pixel space based on the gradient of the objective function to maximize the loss. The other type of attack is called a black-box attack, in which one does not have knowledge of the trained model and would like to generate adversarial data from it. The black-box scenario is more difficult due to the missing knowledge, and one way to solve it is to treat this problem as a black box optimization. To generate an adversarial sample for a given data instance $\boldsymbol{x}^{\dagger} \in \mathbb{R}^{D}$, with $D$ representing the number of pixels in the given image, a common formulation of the adversarial attack is to find a noise sample $\boldsymbol{x}^{*}$ solving

$$\boldsymbol{x}^{*} = \underset{\|\boldsymbol{x}\| \leq \varepsilon}{\operatorname{argmax}} \mathcal{L}\left(g(\boldsymbol{x} + \boldsymbol{x}^{\dagger}), y^{\dagger}\right), \tag{4.3}$$

where $y^{\dagger}$ is the correct label of $\boldsymbol{x}^{\dagger}$, $\mathcal{L}$ is the attack loss function, and $g$ represents the model for attack. Following the adversarial attack paradigm of Carlini & Wagner (2017) and its black-box extension (Chen et al., 2017), we use a soft version of the problem (4.3) as follows:

$$\boldsymbol{x}^{*} = \underset{\boldsymbol{x}}{\arg\min} -\mathrm{CE}\left(g(\boldsymbol{x} + \boldsymbol{x}^{\dagger}), y^{\dagger}\right) + \tilde{\tau}\|\boldsymbol{x}\|^{2},$$

where the cross entropy loss CE is assigned as the attack loss and $g(\cdot)$ outputs the probabilities of different classes, usually using a softmax function for normalization. $\tilde{\tau}$ is a variable balancing the attack loss function

CE and the attack norm. The optimization goal is to find a small shift $x$ in pixel space so that the output results are greatly changed.

In this study, we utilized two convolutional neural network (CNN) architectures to model the HF and LF representations for classification tasks on the MNIST dataset with 60,000 training data and 10,000 testing data. The HF model was a deeper CNN consisting of two convolutional layers, the first with 32 filters and the second with 64 filters, both using $5 \times 5$ kernels, followed by ReLU activations and $2 \times 2$ max-pooling. The flattened output from the convolutional layers ($7 \times 7 \times 64$) was connected to a fully connected layer with 1024 neurons, followed by a 10-class output layer. In contrast, the LF model employed a simplified architecture with a single convolutional layer containing 2 filters and a $3 \times 3$ kernel, followed by ReLU activation and $2 \times 2$ max-pooling. The output ($13 \times 13 \times 2$) was flattened and passed through a fully connected layer with 16 neurons, leading to a 10-class output layer with log-softmax activation. The HF model was designed to provide high-capacity representations, while the LF model served as a lightweight alternative for computational efficiency. The LF model was trained using knowledge distillation (Hinton et al., 2014), leveraging only 1000 training samples and 1000 evaluations of the HF function. Knowledge distillation is a technique where a smaller, simpler model (the student) learns to replicate the outputs of a larger, more complex model (the teacher), effectively transferring knowledge while reducing computational costs. The classification accuracy for the HF and LF CNNs are 99.02% and 82.21%, respectively. There are 27,562 parameters for the LF CNN and 3,274,634 parameters for the HF CNN. We estimate the ratio between HF and LF computational costs as $3274634/27562 \approx 118.8$. The images in MNIST dataset are $28 \times 28$ with a single channel, hence the dimension is $D = 784$. The starting points are initialized as the origin point for all experiments.

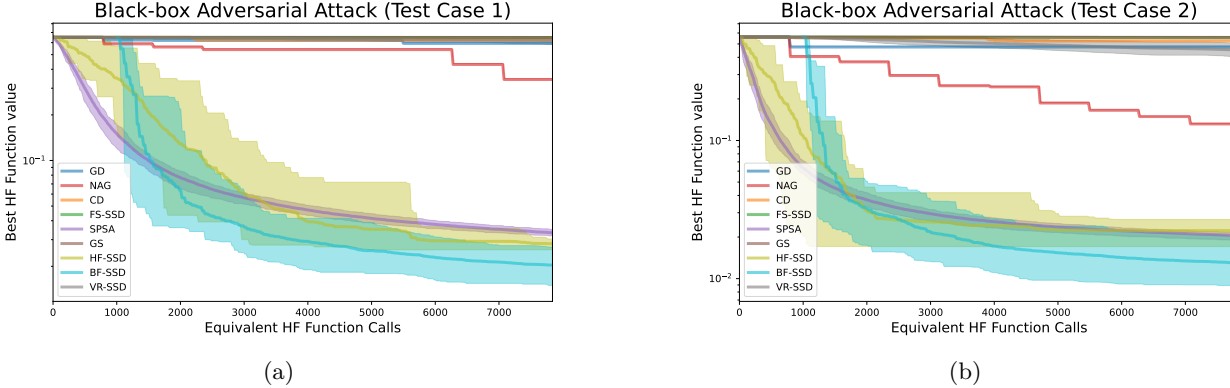

Figure 6: Optimization performances according to different attack targets. The images and their attack noises are presented in Figure 7.

Figure 6 illustrates the convergence of various zeroth-order methods on two test images. For the SSD methods (including the line search version), the parameters are set to $\ell = 50$ and $\alpha_{\max} = 2.0$. Since BF-SSD uses 1,000 HF evaluations for knowledge distillation training, it begins at $N = 1,000$. The convergence results demonstrate that HF-SSD outperforms other methods in this task. Additionally, HF-SSD, BF-SSD, and SPSA exhibit clear advantages over other methods, underscoring the importance of tuning suitable step sizes for the optimization process.

In Figure 7, we present the adversarially attacked test images generated by different optimization approaches for $N = 2,000, 5,000,$ and $7,000$. For the first test image (a-f), only HF-SSD, BF-SSD, and SPSA successfully flip the output of the HF model under limited HF evaluations ($N \leq 7,000$). Similarly, for the second test image (g-l), HF-SSD, BF-SSD, SPSA, and VR-SSD succeed in flipping the HF model output. However, in both cases, we observe that HF-SSD (Figure 7c and Figure 7i) and BF-SSD (Figure 7d and Figure 7j) tend to blur the images more than SPSA (Figure 7b and Figure 7h). This behavior may result from differences in sampling strategies, such as Haar measure sampling versus Hadamard sampling. From an adversarial attack perspective, a successful attack should flip the model's output without excessively blurring the image. In this regard, SPSA outperforms HF-SSD and BF-SSD, although its loss function remains higher than that of

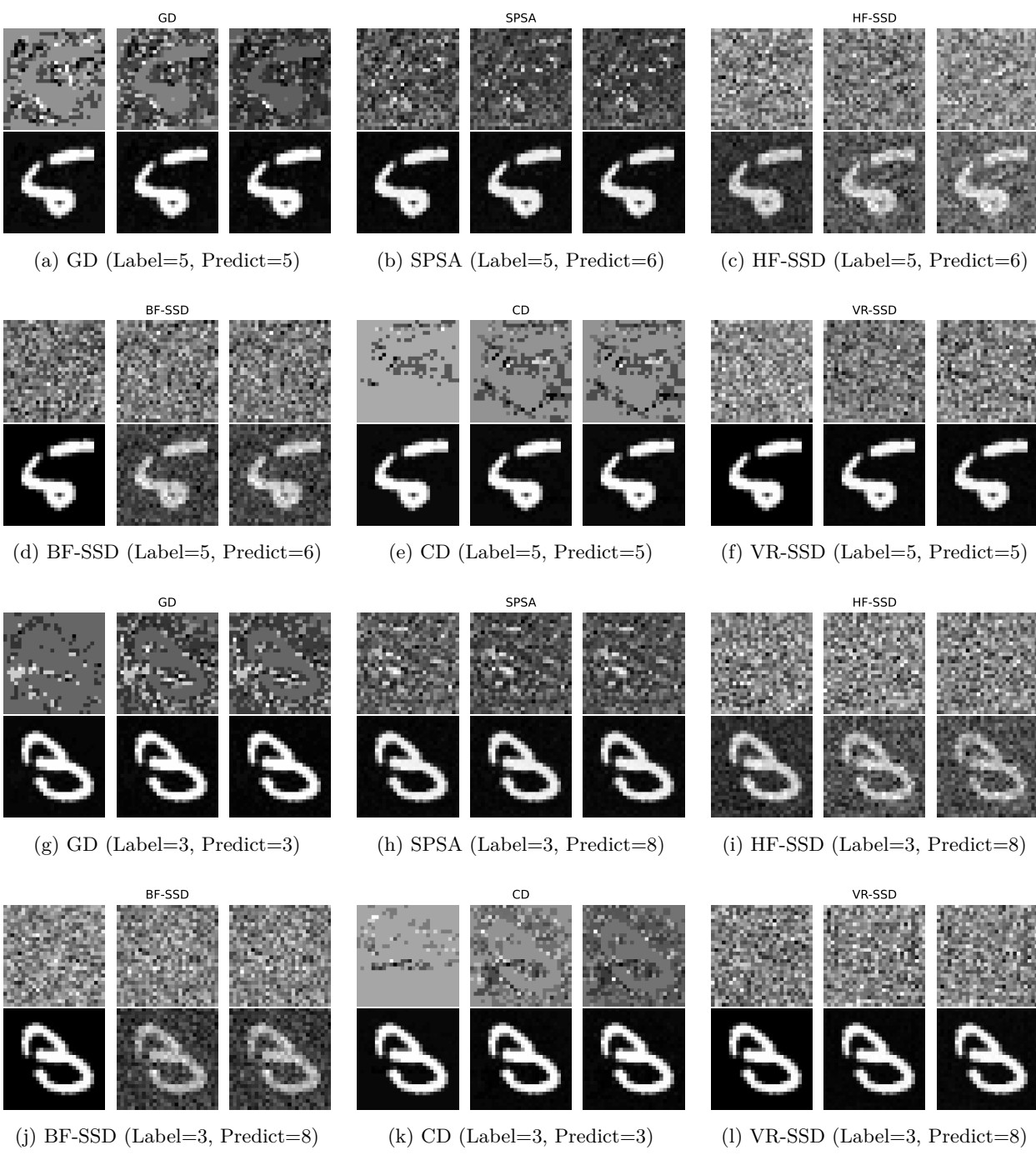

(a) GD (Label=5, Predict=5)   (b) SPSA (Label=5, Predict=6)   (c) HF-SSD (Label=5, Predict=6)

(d) BF-SSD (Label=5, Predict=6)   (e) CD (Label=5, Predict=5)   (f) VR-SSD (Label=5, Predict=5)

(g) GD (Label=3, Predict=3)   (h) SPSA (Label=3, Predict=8)   (i) HF-SSD (Label=3, Predict=8)

(j) BF-SSD (Label=3, Predict=8)   (k) CD (Label=3, Predict=3)   (l) VR-SSD (Label=3, Predict=8)

Figure 7: Adversarial examples for MNIST sample #8 (top two rows) and MNIST sample #18 (bottom two rows) using different methods at $N = 2{,}000, 5{,}000,$ and $7{,}000$.

the other methods. The reason for this observation requires further investigation, but it is out of the scope of this work.

### 4.2.3 Soft Prompting Black-box Language Model

Fine-tuning pre-trained models like BERT or GPT has become a cornerstone of modern natural language processing (NLP). These models, trained on massive corpora, achieve state-of-the-art performance across a wide range of downstream tasks when adapted using task-specific fine-tuning. However, traditional fine-tuning involves updating millions or even billions of parameters, making it computationally expensive and prone to overfitting, especially in low-resource settings. To address these challenges, soft prompting has emerged as a lightweight and efficient alternative. Instead of modifying the model's internal parameters, soft prompting introduces learnable embeddings (soft prompts) that are prepended to the input sequence, enabling task adaptation with minimal computational cost. This approach is particularly appealing for tasks requiring minimal intervention in the model's architecture while leveraging its pre-trained knowledge.

Despite the efficiency of soft prompting, its practical applicability faces challenges when dealing with black-box models where gradients with respect to the model parameters are inaccessible. For instance, many commercial APIs or proprietary models only provide access to predictions or loss values, making gradient-based optimization infeasible. In such scenarios, zeroth-order optimization becomes a crucial tool. Specifically, in this section, we consider a black-box, pre-trained language classifier $f_c : \mathbb{R}^{L_t \times 768} \to [0, 1]$, a pre-trained tokenizer $f_t : \texttt{str} \to \mathbb{R}^{L_t \times 768}$, where $\texttt{str}$ is any string of arbitrary length, and the sequence length $L_t$ is a positive integer up to 512 representing the length of the embedding. The goal is to find a soft prompt $\boldsymbol{x}^* \in \mathbb{R}^{768}$ such that

$$\boldsymbol{x}^* = \arg\min_{\boldsymbol{x}} \mathbb{E}_{(\boldsymbol{z}, y)} \big[ \mathrm{CE}(f_c(\mathrm{cat}[\boldsymbol{x}, f_t(\boldsymbol{z})]), y) \big], \tag{4.4}$$

where $\mathrm{CE}(\cdot, \cdot)$ is the cross-entropy loss function, and the dataset $(\boldsymbol{z}, y) \in \texttt{str} \times \{0, 1\}$. This particular formulation addresses a binary sentiment analysis task where a given string is classified as expressing a positive (1) or negative (0) sentiment. Since the classifier $f_c$ is pre-trained and treated as a black box, gradient information for the loss function is unavailable, necessitating the use of zeroth-order optimization to solve the problem.

For the pre-trained classifier and tokenizer, we employed the BERT model, a state-of-the-art transformer-based architecture trained on large corpora. Specifically, we focused on a simplified version of BERT, named `DistilBERT`. Sentiment analysis on the `aclImdb` dataset was used as a soft prompting task. This dataset comprises movie reviews categorized into positive and negative sentiments, forming a binary classification problem. A $D = 768$ dimension soft prompt $\boldsymbol{x}$ is considered as the input. The transformer's parameters were kept frozen to focus optimization on the soft prompt, reducing the degrees of freedom and computational overhead. For computational convenience, we focus on small scale in this example. The HF model, as described in Equation (4.4), was evaluated using 10 samples from `aclImdb` to approximate the expectation, while the LF model leveraged only 2 samples that were randomly selected from them. Consequently, the evaluation cost ratio between HF and LF was 5:1.

We set the initial starting point at the origin. We let $\ell = 50$, $c = 0.99$, and for the methods without line search we chose a fixed step size of $1 \times 10^{-2}$. The $y$-axis in the following figure represents the average cross-entropy loss. Figure 8 illustrates the performances of different competing methods with the BF-SSD demonstrating notable advantages.

## 5 Conclusion

In this work, we proposed and analyzed a bi-fidelity line search scheme designed to accelerate the convergence of zeroth-order optimization algorithms. By constructing a surrogate model using both computationally expensive high-fidelity (HF) and inexpensive low-fidelity (LF) objective function evaluations, our approach enables an efficient backtracking line search on the surrogate to determine suitable step sizes. This method significantly reduces the reliance on costly HF evaluations, a common bottleneck in zeroth-order methods. We established theoretical convergence guarantees for this scheme under standard assumptions and subsequently integrated it into the stochastic subspace descent framework, yielding the Bi-Fidelity Stochastic Subspace Descent (BF-SSD) algorithm.

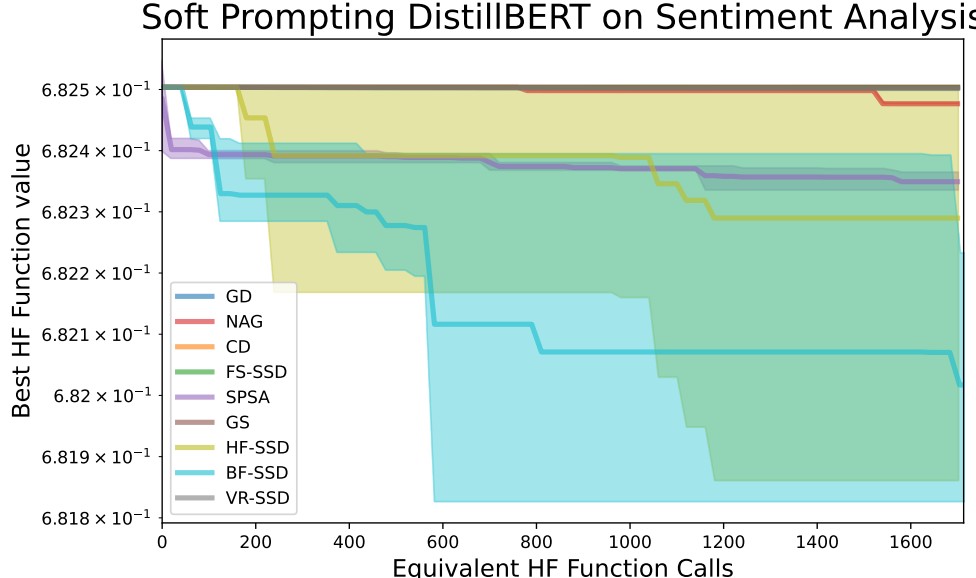

Figure 8: Average cross entropy loss for various zeroth-order optimizers. The BF-SSD method achieves competitive performance while requiring substantially fewer expensive HF function evaluations.

The practical efficacy of BF-SSD was demonstrated through comprehensive experiments across diverse applications, including a synthetic optimization benchmark, dual-form kernel ridge regression, black-box adversarial attacks, and the fine-tuning of transformer-based language models. Across these tasks, BF-SSD consistently outperformed relevant baseline methods, including standard gradient descent, coordinate descent, SPSA, and a purely high-fidelity SSD variant, particularly when comparing solution quality achieved for a given budget of HF evaluations.

These findings underscore the potential of leveraging bi-fidelity information within stochastic subspace methods to effectively address large-scale, high-dimensional optimization problems where function evaluations are expensive. While the current analysis focuses on deterministic subspace steps, future work could extend the theoretical guarantees to encompass the stochastic nature of gradient estimation within the SSD framework. Overall, BF-SSD presents a promising and computationally efficient optimization tool for a variety of challenging real-world applications.

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

## A   Proof of Lemma 2.7

*Proof.* We let $n_k$ evaluations positioned at equispaced points between $\boldsymbol{x}_k$ and $\boldsymbol{x}_k + \alpha_{\max}\boldsymbol{v}_k$, each sub-interval has length $\alpha_{\max}/n_k$. We also define $\psi_k(\alpha) := \varphi(\alpha) - f^{\mathrm{LF}}(\boldsymbol{x} + \alpha\boldsymbol{v}_k)$. For each sub-interval, we define the surrogate $\tilde{\psi}_k(\alpha; n_k)$ as a linear function connecting these values.

WLOG, we prove the bound in Equation (2.4) holds in the interval $\alpha \in [0, h]$ with $h = \alpha_{\max}/n_k$ and this result can be extended to other sub-intervals. The surrogate $\psi(\alpha)$ is defined as

$$\tilde{\psi}_k(\alpha; n_k) := \frac{h - \alpha}{h}\psi(0) + \frac{\alpha}{h}\psi(h), \quad \alpha \in [0, h],$$

and similar definitions of $\tilde{\psi}_k(\alpha; n_k)$ hold when $\alpha$ in other sub-intervals. Such linear approximation $\tilde{\psi}_k(\alpha)$ satisfies

$$|\varphi(\alpha) - \tilde{\varphi}_k(\alpha; n_k)| = |\psi(\alpha) - \tilde{\psi}_k(\alpha; n_k)| = \left|\frac{h - \alpha}{h}\left(\psi(0) - \psi(\alpha)\right) + \frac{\alpha}{h}\left(\psi(h) - \psi(\alpha)\right)\right|, \tag{A.1}$$

for any $\alpha \in [0, h]$. Since the Lipschitz constant of $\psi(\alpha)$ is strictly controlled by $W$ and the fact that $\boldsymbol{v}_k$ is a unit vector, Equation (A.1) satisfies

$$|\varphi(\alpha) - \tilde{\varphi}_k(\alpha; n_k)| \leq W\frac{\alpha_{\max}/n_k - \alpha}{\alpha_{\max}/n_k}(\alpha - 0) + W\frac{\alpha}{\alpha_{\max}/n_k}(\alpha_{\max}/n_k - \alpha) \leq \frac{W\alpha_{\max}}{2n_k}, \quad \forall \alpha \in [0, h].$$

Since we have

$$n_k \geq \frac{WL(1 + c)\alpha_{\max}}{c\beta\|\boldsymbol{v}_k\|^2},$$

the sup-norm is bounded as

$$|\varphi(\alpha) - \tilde{\varphi}_k(\alpha; n_k)| \leq \frac{c\beta\|\boldsymbol{v}_k\|^2}{2(1 + c)L} = \frac{\|\boldsymbol{v}_k\|^2}{2}\min\left\{\frac{c}{(1 + c)^2 L}, \frac{c\beta}{(1 + c)L}, \beta\alpha_{\max}\right\},$$

where the last equality stems from the fact that $\beta \leq 1/2$ and $\alpha_{\max} \geq c/(cL + L)$. □

## B   Single-fidelity SSD with Line Search

### B.1   Assuming Strong-convexity

**Assumption B.1.** Assume the objective function $f^{\mathrm{HF}}$ and algorithm satisfies the following conditions

1. $\boldsymbol{P}_k \in \mathbb{R}^{D \times \ell}$ are independent random matrices such that $\mathbb{E}[\boldsymbol{P}_k\boldsymbol{P}_k^T] = \boldsymbol{I}_d$ and $\boldsymbol{P}_k^T\boldsymbol{P}_k = (D/\ell)\boldsymbol{I}_\ell$ with $D > \ell$;

2. Objective function $f^{\mathrm{HF}} : \mathbb{R}^D \to \mathbb{R}$ attains its minimum $f^*$ and $\nabla f^{\mathrm{HF}}$ is $L$-Lipschitz continuous;

3. Objective function $f^{\mathrm{HF}} : \mathbb{R}^D \to \mathbb{R}$ is $\gamma$-strongly convex; note $\gamma \leq L$.

**Theorem B.2.** *(Single fidelity) With the assumptions of B.1, SSD with line search (either exact line search or backtracking) converges in the sense that $f(\boldsymbol{x}_k) \overset{a.s.}{\to} f^*$ and $f(\boldsymbol{x}_k) \overset{L^1}{\to} f^*$.*

*In particular,*

$$\mathbb{E}[f(\boldsymbol{x}_{k+1})] - f^* \leq \omega^{k+1}(f(\boldsymbol{x}_0) - f^*),$$

*for $\omega \in (0, 1)$ where*

$$\omega = \begin{cases} 1 - \frac{\gamma\ell}{DL} & \text{exact line search} \\ 1 - \min\left\{2\gamma\beta\alpha_{\max}, \frac{2\ell c\gamma\beta}{DL}\right\} & \text{backtracking.} \end{cases}$$

*Proof.* Define the filtration $\mathcal{F}_k \coloneqq \sigma(\boldsymbol{P}_1, \ldots, \boldsymbol{P}_{k-1})$ and $\mathcal{F}_1 = \{\emptyset, \Omega_P\}$, with $\Omega_P$ as the sample space. By Lipschitz continuity, we have

$$f(\boldsymbol{x}_{k+1}) \leq f(\boldsymbol{x}_k) + \nabla f(\boldsymbol{x}_k)^T (\boldsymbol{x}_{k+1} - \boldsymbol{x}_k) + \frac{L}{2} \|\boldsymbol{x}_{k+1} - \boldsymbol{x}_k\|^2. \tag{B.1}$$

By defining $f_e(\boldsymbol{x}) \coloneqq f(\boldsymbol{x}) - f^*$ and plugging $\boldsymbol{x}_{k+1} = \boldsymbol{x}_k - \alpha_k \boldsymbol{P}_k \boldsymbol{P}_k^T \nabla f(\boldsymbol{x}_k)$, Equation (B.1) yields

$$
\begin{aligned}
f_e(\boldsymbol{x}_{k+1}) - f_e(\boldsymbol{x}_k) &\leq -\alpha_k \langle \nabla f(\boldsymbol{x}_k), \boldsymbol{P}_k \boldsymbol{P}_k^T \nabla f(\boldsymbol{x}_k) \rangle + \frac{\alpha_k^2 L}{2} \langle \boldsymbol{P}_k \boldsymbol{P}_k^T \nabla f(\boldsymbol{x}_k), \boldsymbol{P}_k \boldsymbol{P}_k^T \nabla f(\boldsymbol{x}_k) \rangle \\
&= -\alpha_k \langle \nabla f(\boldsymbol{x}_k), \boldsymbol{P}_k \boldsymbol{P}_k^T \nabla f(\boldsymbol{x}_k) \rangle + \frac{D\alpha_k^2 L}{2\ell} \langle \nabla f(\boldsymbol{x}_k), \boldsymbol{P}_k \boldsymbol{P}_k^T \nabla f(\boldsymbol{x}_k) \rangle \\
&= \left( -\alpha_k + \frac{D\alpha_k^2 L}{2\ell} \right) \langle \nabla f(\boldsymbol{x}_k), \boldsymbol{P}_k \boldsymbol{P}_k^T \nabla f(\boldsymbol{x}_k) \rangle,
\end{aligned} \tag{B.2}
$$

where the fact $\boldsymbol{P}_k \boldsymbol{P}_k^T \boldsymbol{P}_k \boldsymbol{P}_k^T = (D/\ell) \boldsymbol{P}_k \boldsymbol{P}_k^T$ is applied. We have two line search approaches to determine the step size $\alpha_k$,

1. Exact line search:
$$\alpha_k = \arg\min_{\alpha} f(\boldsymbol{x}_k - \alpha \boldsymbol{P}_k \boldsymbol{P}_k^T \nabla f(\boldsymbol{x}_k)); \tag{B.3}$$

2. Backtracking: for some fixed $\alpha_{\max} > 0$, $\beta \in (0, \ell/2d)$, and $c \in (0,1)$,
$$\alpha_k = \max_{m \in \mathbb{N}} c^m \alpha_{\max}$$
$$\text{s.t.} \quad f(\boldsymbol{x}_k - c^m \alpha_{\max} \boldsymbol{P}_k \boldsymbol{P}_k^T \nabla f(\boldsymbol{x}_k)) \leq f(\boldsymbol{x}^k) - \beta c^m \alpha_{\max} \|\boldsymbol{P}_k \boldsymbol{P}_k^T \nabla f(\boldsymbol{x}^k)\|^2. \tag{B.4}$$

We will prove the convergence for two line search methods separately. All the following analyses hold for any $\boldsymbol{P}_k$ satisfying Assumption B.1.

**Exact line search** According to Equation (B.3), the exact line search method can find the optimal $\alpha_k$ such that the quadratic term in Equation (B.2) yields $-\alpha_k + D\alpha_k^2 L/2\ell \leq -\ell/(2dL)$ for any $\boldsymbol{P}_k$, thereby

$$f_e(\boldsymbol{x}_{k+1}) - f_e(\boldsymbol{x}_k) \leq -\frac{\ell}{2dL} \langle \nabla f(\boldsymbol{x}_k), \boldsymbol{P}_k \boldsymbol{P}_k^T \nabla f(\boldsymbol{x}_k) \rangle \quad \forall \boldsymbol{P}_k.$$

With condition on the current filtration $\mathcal{F}_k$, the conditional expectation on both sides turn to

$$
\begin{aligned}
\mathbb{E}[f_e(\boldsymbol{x}_{k+1})|\mathcal{F}_k] &\leq -\frac{\ell}{2dL} \mathbb{E}\left[ \langle \nabla f(\boldsymbol{x}_k), \boldsymbol{P}_k \boldsymbol{P}_k^T \nabla f(\boldsymbol{x}_k) \rangle | \mathcal{F}_k \right] + f_e(\boldsymbol{x}_k) \\
&= -\frac{\ell}{2dL} \|\nabla f(\boldsymbol{x}_k)\|^2 + f_e(\boldsymbol{x}_k),
\end{aligned} \tag{B.5}
$$

where the equality is from the fact $\mathbb{E}[\boldsymbol{P}_k \boldsymbol{P}_k^T | \mathcal{F}_k] = \boldsymbol{I}_d$. By invoking the Polyak-Lojasiewicz inequality,

$$\mathbb{E}[f_e(\boldsymbol{x}_{k+1})|\mathcal{F}_k] \leq -\frac{\gamma\ell}{DL} f_e(\boldsymbol{x}_k) + f_e(\boldsymbol{x}_k) = \left( 1 - \frac{\gamma\ell}{DL} \right) f_e(\boldsymbol{x}_k). \tag{B.6}$$

Recursive application yields

$$\mathbb{E}[f_e(\boldsymbol{x}_{k+1})|\mathcal{F}_k] \leq \left( 1 - \frac{\gamma\ell}{DL} \right) f_e(\boldsymbol{x}_k) = \left( 1 - \frac{\gamma\ell}{DL} \right) \mathbb{E}[f_e(\boldsymbol{x}_k)|\mathcal{F}_{k-1}] \leq \left( 1 - \frac{\gamma\ell}{DL} \right)^{k+1} \mathbb{E}[f_e(\boldsymbol{x}_0)]. \tag{B.7}$$

Since $\ell \leq D$ and $\gamma \leq L$, the term $1 - \gamma\ell/DL$ is less than 1. Equation (B.7) implies

$$\mathbb{E}[f(\boldsymbol{x}_{k+1})] - f^* \leq \left( 1 - \frac{\gamma\ell}{DL} \right)^{k+1} \mathbb{E}[(f(\boldsymbol{x}_0) - f^*)] = \left( 1 - \frac{\gamma\ell}{DL} \right)^{k+1} (f(\boldsymbol{x}_0) - f^*),$$

which proves $f(\boldsymbol{x}_k) \overset{a.s.}{\to} f^*$ and $f(\boldsymbol{x}_k) \overset{L^1}{\to} f^*$.

**Backtracking** (Showing the existence of a feasible set such that the Armijo condition is satisfied.) Following Equation (B.4), the backtracking method selects the maximal possible step size value that satisfies the Armijo condition with specified parameter $\beta \leq \ell/2d$ and shrinking parameter $c < 1$. When $0 \leq \alpha_k \leq \ell/DL$, $-\alpha_k + D\alpha_k^2 L/2\ell \leq -\alpha_k/2$ holds with Haar measure probability one, which implies the following Armijo stopping condition

$$
\begin{aligned}
f(\boldsymbol{x}_{k+1}) &\leq f(\boldsymbol{x}_k) - \alpha_k \langle \nabla f(\boldsymbol{x}_k), \boldsymbol{P}_k \boldsymbol{P}_k^T \nabla f(\boldsymbol{x}_k) \rangle + \frac{D\alpha_k^2 L}{2\ell} \langle \nabla f(\boldsymbol{x}_k), \boldsymbol{P}_k \boldsymbol{P}_k^T \nabla f(\boldsymbol{x}_k) \rangle \\
&\leq f(\boldsymbol{x}_k) - \frac{\alpha_k}{2} \langle \nabla f(\boldsymbol{x}_k), \boldsymbol{P}_k \boldsymbol{P}_k^T \nabla f(\boldsymbol{x}_k) \rangle \\
&= f(\boldsymbol{x}_k) - \frac{\alpha_k \ell}{2d} \| \boldsymbol{P}_k \boldsymbol{P}_k^T \nabla f(\boldsymbol{x}_k) \|^2 \\
&\leq f(\boldsymbol{x}_k) - \beta \alpha_k \| \boldsymbol{P}_k \boldsymbol{P}_k^T \nabla f(\boldsymbol{x}_k) \|^2.
\end{aligned}
$$

Therefore, the backtracking terminates when $\alpha_k = \alpha_{\max}$ or $\alpha_k \geq \ell c/DL$, which implies

$$
f_e(\boldsymbol{x}_{k+1}) \leq f_e(\boldsymbol{x}_k) - \beta \alpha_{\max} \langle \nabla f(\boldsymbol{x}_k), \boldsymbol{P}_k \boldsymbol{P}_k^T \nabla f(\boldsymbol{x}_k) \rangle, \tag{B.8}
$$

or

$$
f_e(\boldsymbol{x}_{k+1}) \leq f_e(\boldsymbol{x}_k) - \frac{\ell c \beta}{DL} \langle \nabla f(\boldsymbol{x}_k), \boldsymbol{P}_k \boldsymbol{P}_k^T \nabla f(\boldsymbol{x}_k) \rangle. \tag{B.9}
$$

Similar with Equation (B.5), by combining Equations (B.8) and (B.9), and taking expectations conditioned on the filtration $\mathcal{F}_k$, we have

$$
\begin{aligned}
\mathbb{E}[f_e(\boldsymbol{x}_{k+1})|\mathcal{F}_k] &\leq -\min\left\{\beta \alpha_{\max}, \frac{\ell c \beta}{DL}\right\} \mathbb{E}\left[\langle \nabla f(\boldsymbol{x}_k), \boldsymbol{P}_k \boldsymbol{P}_k^T \nabla f(\boldsymbol{x}_k) \rangle | \mathcal{F}_k\right] + f_e(\boldsymbol{x}_k) \\
&= -\min\left\{\beta \alpha_{\max}, \frac{\ell c \beta}{DL}\right\} \|\nabla f(\boldsymbol{x}_k)\|^2 + f_e(\boldsymbol{x}_k).
\end{aligned} \tag{B.10}
$$

Similar to Equation (B.6), by invoking the Polyak-Lojasiewicz inequality,

$$
\mathbb{E}[f_e(\boldsymbol{x}_{k+1})|\mathcal{F}_k] \leq -\min\left\{2\gamma\beta\alpha_{\max}, \frac{2\ell c \gamma \beta}{DL}\right\} f_e(\boldsymbol{x}_k) + f_e(\boldsymbol{x}) = \left(1 - \min\left\{2\gamma\beta\alpha_{\max}, \frac{2\ell c \gamma \beta}{DL}\right\}\right) f_e(\boldsymbol{x}_k). \tag{B.11}
$$

By recursively implementing Equation (B.11), we have

$$
\mathbb{E}[f_e(\boldsymbol{x}_{k+1})] \leq \left(1 - \min\left\{2\gamma\beta\alpha_{\max}, \frac{2\ell c \gamma \beta}{DL}\right\}\right)^{k+1} f_e(\boldsymbol{x}_0).
$$

Since $0 < c < 0$, $0 < \gamma \leq L$, $0 < \ell \leq D$, and $0 < \beta < 0.5$, the term $0 < (1 - \min\{2\gamma\beta, 2\ell c\gamma\beta/DL\}) < 1$, the convergences $f(\boldsymbol{x}_k) \overset{a.s.}{\to} f^*$ and $f(\boldsymbol{x}_k) \overset{L^1}{\to} f^*$ are guaranteed. $\qquad\square$

## B.2 Assuming Convexity

**Assumption B.3.** For the non-strongly convex objective function $f^{\mathrm{HF}}$ and the algorithm, we make the following assumptions

1. $\boldsymbol{P}_k \in \mathbb{R}^{D \times \ell}$ are independent random matrices such that $\mathbb{E}[\boldsymbol{P}_k \boldsymbol{P}_k^T] = \boldsymbol{I}_d$ and $\boldsymbol{P}_k^T \boldsymbol{P}_k = (D/\ell)\boldsymbol{I}_\ell$ with $D > \ell$;

2. Objective function $f^{\mathrm{HF}} : \mathbb{R}^D \to \mathbb{R}$ is convex and $\nabla f^{\mathrm{HF}}$ is $L$-Lipschitz continuous;

3. The function $f^{\mathrm{HF}}$ attains its minimum(s) $f^*$ at $x^*$ so that there exists a known $R$ satisfying $\max_{\boldsymbol{x},\boldsymbol{x}^*}\{\|\boldsymbol{x} - \boldsymbol{x}^*\| : f^{\mathrm{HF}}(\boldsymbol{x}) \leq f^{\mathrm{HF}}(\boldsymbol{x}_0)\} \leq R$;

Given the above convex-but-not-strongly-convex assumption, we have the following $L^1$ convergence result:

**Theorem B.4.** *With backtracking implemented for line search, we have*

$$\mathbb{E}[f(\boldsymbol{x}_k)] - f^* \leq \max\left\{\frac{2R^2}{k\beta\alpha_{\max}}, \frac{2dLR^2}{k\ell c\beta}\right\}.$$

*Proof.* We use the same notation as in the proof of Thm. B.2. Starting from Equation (B.10), we have

$$\mathbb{E}[f_e(\boldsymbol{x}_{k+1})|\mathcal{F}_k] = -\min\left\{\beta\alpha_{\max}, \frac{\ell c\beta}{DL}\right\}\|\nabla f(\boldsymbol{x}_k)\|^2 + f_e(\boldsymbol{x}_k).$$

By convexity and the Cauchy-Schwartz inequality, $\|\nabla f(\boldsymbol{x}_k)\| \geq f_e(\boldsymbol{x}_k)/R$. With the fact that $\mathbb{E}f_e(\boldsymbol{x}_{k+1}) \leq \mathbb{E}f_e(\boldsymbol{x}_k)$, we have

$$\begin{aligned}
\mathbb{E}f_e(\boldsymbol{x}_{k+1}) - f_e(\boldsymbol{x}_k) &\leq -\min\left\{\beta\alpha_{\max}, \frac{\ell c\beta}{DL}\right\}\frac{\mathbb{E}f_e^2(\boldsymbol{x}_k)}{2R^2}\\
&\leq -\min\left\{\beta\alpha_{\max}, \frac{\ell c\beta}{DL}\right\}\frac{\mathbb{E}^2 f_e(\boldsymbol{x}_k)}{2R^2}\\
&\leq -\min\left\{\beta\alpha_{\max}, \frac{\ell c\beta}{DL}\right\}\frac{\mathbb{E}f_e(\boldsymbol{x}_{k+1})\mathbb{E}f_e(\boldsymbol{x}_k)}{2R^2},
\end{aligned}$$

which further implies

$$\frac{1}{\mathbb{E}f_e(\boldsymbol{x}_{k+1})} \geq \frac{1}{\mathbb{E}f_e(\boldsymbol{x}_k)} + 2R^2\min\left\{\beta\alpha_{\max}, \frac{\ell c\beta}{DL}\right\}. \tag{B.12}$$

Applying (B.12) recursively, we obtain

$$\mathbb{E}f_e(\boldsymbol{x}_{k+1}) \leq \left(\min\left\{\beta\alpha_{\max}, \frac{\ell c\beta}{DL}\right\}\right)^{-1}\frac{2R^2}{k} = \max\left\{\frac{2R^2}{k\beta\alpha_{\max}}, \frac{2dLR^2}{k\ell c\beta}\right\}.$$

$\square$

### B.3 No convexity assumptions

**Assumption B.5.** We make the following assumptions:

1. $\boldsymbol{P}_k \in \mathbb{R}^{D\times\ell}$ are independent random matrices such that $\mathbb{E}[\boldsymbol{P}_k\boldsymbol{P}_k^T] = \boldsymbol{I}_d$ and $\boldsymbol{P}_k^T\boldsymbol{P}_k = (D/\ell)\boldsymbol{I}_\ell$ with $D > \ell$;

2. The objective function $f^{\mathrm{HF}} : \mathbb{R}^D \to \mathbb{R}$ (or $f^{\mathrm{HF}}$) attains its minimum $f^*$ and $\nabla f^{\mathrm{HF}}$ is $L$-Lipschitz continuous;

When Assumption B.5 holds, we have the following $L^2$ convergence of the gradient norm result for SSD with line search:

**Theorem B.6** (Same as Theorem 3.1)**.** *With Assumption B.5 holding and backtracking implemented for line search, we have*

$$\min_{k\in\{0,\dots,K\}}\mathbb{E}[\|\nabla f(\boldsymbol{x}_k)\|^2] \leq \max\left\{\frac{(f(\boldsymbol{x}_0) - f^*)}{(K+1)\beta\alpha_{\max}}, \frac{DL(f(\boldsymbol{x}_0) - f^*)}{(K+1)\ell c\beta}\right\}.$$

*That is, $k = \mathcal{O}(1/(\epsilon\beta\alpha_{\max}) + DL/(\epsilon\ell c\beta))$ iterations are required to achieve $\mathbb{E}\|\nabla f(\boldsymbol{x}_k)\|^2 \leq \epsilon$.*

*Proof.* Following Equation (B.10)

$$\min\left\{\beta\alpha_{\max}, \frac{\ell c\beta}{DL}\right\}\|\nabla f(\boldsymbol{x}_k)\|^2 \leq f_e(\boldsymbol{x}_k) - \mathbb{E}[f_e(\boldsymbol{x}_{k+1})|\mathcal{F}_k],$$

which leads to the telescope series

$$\min\left\{\beta\alpha_{\max}, \frac{\ell c\beta}{DL}\right\}\sum_{k=0}^{K}\|\nabla f(\boldsymbol{x}_k)\|^2 \leq \sum_{k=0}^{K}\left(f_e(\boldsymbol{x}_k) - \mathbb{E}[f_e(\boldsymbol{x}_{k+1})|\mathcal{F}_k]\right)$$
$$= f(\boldsymbol{x}_0) - \mathbb{E}f(\boldsymbol{x}_{K+1}) \leq f(\boldsymbol{x}_0) - f^*.$$

Therefore,

$$(K+1)\min_{k\in\{0,\dots,K\}}\mathbb{E}\|\nabla f(\boldsymbol{x}_k)\|^2 \leq \left(\min\left\{\beta\alpha_{\max}, \frac{\ell c\beta}{DL}\right\}\right)^{-1}(f(\boldsymbol{x}_0) - f^*)$$
$$= \max\left\{\frac{(f(\boldsymbol{x}_0) - f^*)}{\beta\alpha_{\max}}, \frac{DL(f(\boldsymbol{x}_0) - f^*)}{\ell c\beta}\right\}.$$

A sufficient condition to let $\mathbb{E}\|\nabla f(\boldsymbol{x}_k)\|^2$ be $\epsilon$-small is to let

$$k \geq \max\left\{\frac{(f(\boldsymbol{x}_0) - f^*)}{\epsilon\beta\alpha_{\max}}, \frac{DL(f(\boldsymbol{x}_0) - f^*)}{\epsilon\ell c\beta}\right\}.$$

$\square$

## C   Worst Function in the World: Additional Data

| Method | $c = 0.8$ | | | $c = 0.9$ | | | $c = 0.99$ | | |
|--------|-----------|-----------|-----------|-----------|-----------|-----------|-----------|-----------|-----------|
|        | $\ell = 5$ | $\ell = 10$ | $\ell = 20$ | $\ell = 5$ | $\ell = 10$ | $\ell = 20$ | $\ell = 5$ | $\ell = 10$ | $\ell = 20$ |
| GD     | 0.9026 | 0.9026 | 0.9026 | 0.9026 | 0.9026 | 0.9026 | 0.9026 | 0.9026 | **0.9026** |
| CD     | 0.8984 | 0.8984 | 0.8984 | 0.8984 | 0.8984 | 0.8984 | 0.8984 | 0.8984 | **0.8984** |
| FS-SSD | 2.4495 | 2.4194 | **2.3611** | 2.4495 | 2.4196 | 2.3619 | 2.4497 | 2.4194 | 2.3622 |
| SPSA   | 0.7713 | 0.7756 | 0.7502 | 0.6245 | **0.4623** | 0.5549 | 0.6046 | 0.6721 | 0.7006 |
| GS     | 2.4598 | 2.4442 | **2.4129** | 2.4597 | 2.4447 | 2.4144 | 2.4596 | 2.4445 | 2.4150 |
| HF-SSD | 0.3620 | 0.2511 | **0.2194** | 0.8667 | 0.5109 | 0.3337 | 3.4582 | 1.7191 | 1.0331 |
| BF-SSD | 0.3177 | 0.2947 | 0.2932 | 0.2686 | 0.2526 | 0.2497 | 0.2316 | 0.2104 | **0.1984** |
| VR-SSD | 0.9885 | 0.9374 | 0.9178 | 0.9881 | 0.9464 | 0.9154 | 0.9925 | 0.9395 | **0.9121** |

Table 4: Performance values for different optimization methods across various $c$ and $\ell$ combinations at $N = 5,000$. The minimum value in each row is highlighted in bold.

