# OpenReview forum: "Stochastic Subspace Descent Accelerated via Bi-fidelity Line Search"
_TMLR — Accepted by TMLR_

### Review · Reviewer_PMqd · 2025-05-23

**Summary Of Contributions:**

This paper proposes a new algorithm, Bi-Fidelity Stochastic Subspace Descent (BF-SSD), for high-dimensional 0-th-order optimization where evaluating the objective function is expensive and gradients are unavailable.

The paper introduces a line search method that leverages a combination of cheap low-fidelity (LF) evaluations and a small number of expensive high-fidelity (HF) queries to build a simple 1-dimensional surrogate along the search direction. This surrogate is used to adaptively choose the step size, reducing HF query cost per iteration. The method is combined with Stochastic Subspace Descent (SSD) — a zeroth-order technique that estimates descent directions in randomly sampled low-dimensional subspaces — to form the full BF-SSD algorithm.

The paper proves theoretical convergence guarantees under the assumption that the descent direction is accurate (i.e., equivalent to the true gradient), and that the difference between HF and LF models is Lipschitz continuous.

The algorithm is tested on four tasks to support improved performance in terms of HF query efficiency compared to several zeroth-order baselines.

**Audience:**

Yes

**Claims And Evidence:**

Yes

**Requested Changes:**

All below changes are needed for sufficient novelty for the derivative-free community:
(1) Please extend the experiment to higher dimensions, it is not clear to the reader how well the LF surrogate extends to higher dimensional input space. The mismatch/mis-align between LF and HF models will be more serious in high dimensions, and it will be helpful to see if BF-SSD can actually works for standard high-dimensional optimization (around 100-1000 dimensional functions).

(2)It is interesting to link the method in the context of building suitable LF models in section 2.4; but in practice, defining an appropriate low-fidelity function LF​ is non-trivial. The paper assumes you can just “construct” an LF model that is aligned (Assumption 2.2.): but doesn’t provide a metric for LF-HF alignment; I think at least an assessment of failure modes when LF is poorly correlated with HF should be provided.

(3)Table 3 reports the comparison between HF-SSD and BF-SSD, however, HF-SSD does not have similar performance as BF-SSD, which is counter-intuitive. I have also tried to understand the npz file provided in their supplementary files, yet I cannot reproduce the results reported in this table. Could the authors explain why we should expect HF (high-fidelity) is much worse than bi-fidelity? Should we have more information from HF case?

(4)The abstract calls the approach “computationally efficient for large-scale problems”, yet no experiment demonstrates scale nor reports runtime.  LF models themselves are not free: training the distillation network uses 1000 HF calls upfront , but this is simply tacked onto the first iteration and then ignored in plots. Likewise, Nyström preprocessing for kernel ridge regression is not counted. Wall-clock time or GPU hours are never reported. Also it may be more convincing to have a larger scale of ML tasks, in transformer fine-tuning, they only used DistilBERT + 10 samples, which is hardly representative of a real black-box setting.

**Strengths And Weaknesses:**

Strength:
Expensive black-box optimisation is indeed a pain-point in ML and engineering. A method that claims to reduce costly HF queries is automatically interesting.

The SSD samples a random low-dimensional subspace (dimension \ell<<D), then it estimates the gradient projected onto that subspace using finite differences. The author argues that along such a descent direction v_k it costs less computational resource, since the 1D surrogate is a piecewise linear function with few points and a scalar blending with LF; its main difference from traditional surrogate-based 0-th order methods is that rather than modeling the full objective BF-SSD just models the function along the line only; this shares the spirit of arxiv:1301.1942 and is not new in the community. And as the authors acknowledge, SSD that approximates the gradient using finite differences in a random low-dimensional subspace. The integration of bi-fidelity modeling with stochastic subspace descent for zeroth-order optimization is a relatively fresh idea.

Despite length, the narrative is mostly fluent and figures are legible (some fonts like Figure 3 legend could be larger, and use the same legend).

Weaknesses:
However, the idea of using a LF along a low-dimensional space is questionable; In addition, line search on a crude 1D surrogate may not generalize well across problem types, especially with nonlinear LF-HF mismatch; yet the paper provides convergence bounds under technical assumptions (smoothness of the HF objective and LF–HF alignment). It is not clear how this bound helps us to understand the advantage of bi-fidelity modeling better. What is a bit un-usual is to assume that we have access to almost exact gradient v_k's, I understand this is a technical assumption (Assumption 2.3), yet I have to say that this looks overly optimistic for *most* 0-th order applications. To give an example, it is hard to believe that LLM example would satisfy this..or it is a reasonable assumption to make for it.

In fact, the complexity bound depends on an unknown Lipschitz difference W between HF and scaled LF functions (Assumptions 2.2–2.3) . No guidance is given on estimating W or choosing n_k​; in experiments they just set n_k=1 “for simplicity” , undermining the theoretical premises.

While the idea of using bi-fidelity models for step size tuning is novel in the context of SSD, the surrogate model is rudimentary and the analysis applies only under strong assumptions. The empirical results are encouraging but limited in scale and scope.
I am not sure how practical this proposed method is based on the current manuscript.

---

> ### Author Response · Authors · 2025-08-15
>
> > Please extend the experiment ... for standard high-dimensional optimization (around 100-1000 dimensional functions).
>
> We think there was confusion here.  We agree that high-dimensional optimization “around 100-1000 dimensional” is interesting… but our experiments were high dimensional by this definition!  In particular:
>
> * 4.2.1 kernel ridge regression: this is 1000 dimensional
> * 4.2.2 attack on MNIST: this is 784 dimensional
> * 4.2.3 soft prompting: this is 768 dimensional
>
> > It is interesting to link the method ... when LF is poorly correlated with HF should be provided.
>
> While our analysis assumes we have a LF model, we also explicitly give examples, both generic mathematical examples in section 2.4 and very explicit examples in section 4.2. Some of these are generic enough that they can give ideas to other researchers on how to construct their own LF models.  In practice, we found that these simple LF models worked very well.
>
> We do agree it would be great to have an adaptive mechanism to detect poorly correlated models, but feel that’s beyond the scope of the paper.  We also note that one could perhaps do a comparison of predicted vs actual improvement, similar to what we suggested to the other reviewer asking about constructing the surrogate.
>
> > Table 3 reports the comparison ... more information from HF case?
>
> We have updated Tables 1-3 to reflect the most recent results.
>
> The performance difference you noted is indeed the central trade-off our paper explores. The reason BF-SSD outperforms HF-SSD is rooted in how each method determines its step size:
>
> * HF-SSD performs an expensive line search that requires many HF function calls to find the optimal step size.
>
> * BF-SSD, in contrast, uses only a single HF call per iteration, leveraging numerous cheap LF calls to efficiently approximate the line search.
>
> Therefore, in scenarios where LF calls are significantly cheaper than HF calls (as is the case in our experiments), BF-SSD becomes far more cost-effective. This principle is also visually demonstrated in Figure 2 of the paper, which shows BF-SSD making much faster progress for the same computational budget.
>
> We regret that the initial supplementary files were unclear. We have uploaded a new version with two Jupyter notebooks, reproduce_figures.ipynb and reproduce_tables.ipynb. Due to the supplementary size limit, we remove some of the data, but you could generate them by calling the corresponding python files and visualize the results.
>
> > The abstract calls ... a real black-box setting.
>
> 1.  On the Use of "Equivalent HF Calls" as the Primary Metric:
> Our use of "equivalent number of high-fidelity (HF) oracle calls" as the primary cost metric is rooted in its widespread adoption as a standard in the black-box optimization and multi-fidelity modeling literature [1, 2]. This metric provides a clear, implementation-independent basis for comparing algorithm performance, focusing on what is typically the most significant bottleneck: the expensive HF function evaluations. In practical applications where the HF function dominates the overall runtime, the overhead from other parts of the optimization process is negligible.
>
> [1] David Kozak, Stephen Becker, Alireza Doostan, and Luis Tenorio. A stochastic subspace approach to gradient-free optimization in high dimensions. Computational Optimization and Applications, 79(2):339–368, 2021.
>
> [2] Leo W. T. Ng and Karen E. Willcox. Multifidelity approaches for optimization under uncertainty. International Journal for numerical methods in Engineering, 100(10):746–772, 2014.
>
> 2.  On Wall-Clock Time and Upfront Costs:
> To directly validate this assumption, we have conducted a new set of experiments measuring performance against wall-clock time. The [results](https://limewire.com/d/cgXQn#LrFI7AyFC4), which are available for your review here, show a strong alignment with the equivalent HF call data. The relative performance of all methods remains consistent, confirming that the conclusions of our paper are robust to the choice of metric.
>
> We also acknowledge the need to account for upfront costs:
>
> * Distillation Network: We have updated our plots to properly account for the initial 1000 HF calls used for training. This cost is now reflected by shifting the BF-SSD curve to the right, ensuring a fair comparison.
>
> * Nyström Preprocessing: We maintain that the cost of sampling for the Nyström approximation is negligible compared to the expense of the HF calls in the kernel ridge regression experiment.
>
> 3.  On the Scale of ML Tasks:
> We appreciate your feedback on the scale of the transformer fine-tuning task. Our experiment, using DistilBERT with 10 samples, was designed as a proof-of-concept to demonstrate the method's applicability. We agree that evaluating on larger-scale ML tasks is a valuable direction for future work.

---

### Review · Reviewer_6TiG · 2025-07-15

**Summary Of Contributions:**

In this submission, the authors introduced a bi-fidelity line search scheme tailored for zeroth-order optimization. This method constructed a temporary surrogate model by strategically combining inexpensive low-fidelity (LF) evaluations with accurate high-fidelity (HF) evaluations of the objective function. The authors proved theoretical convergence rates of this bi-fidelity line search scheme under standard assumptions. Moreover, the authors integrated this bi-fidelity strategy into the stochastic subspace descent framework and proposed the bi-fidelity stochastic subspace descent (BF-SSD) algorithm. The authors proved the convergence rates of this novel algorithm under non-convex condition, convex condition and strongly convex condition. The authors also conducted comprehensive numerical experiments on the one synthetic function and three practical applications.

**Audience:**

Yes

**Broader Impact Concerns:**

No Broader Impact Concerns

**Claims And Evidence:**

Yes

**Requested Changes:**

As summarized in the weakness section, this submission need the following changes before accepted as publication:

1. The authors need to explain more details about Assumption 2.3. In section 2.4, the authors should explain more about how those concrete function examples satisfy the Assumption 2.3.

2. The authors need to move the convergence rates of SSD with line search from Appendix B to the main part of the submission. The authors also need to change the order: first introduce the results of non-convexity, then the results of convexity and finally the results of strong convexity.

3. Please supplement the performance of accelerated gradient descent and compare it with all other zeroth-order optimization methods in the numerical experiments.

4. The authors should add the linear convergence rates in the strongly convexity condition in Theorem B.2. instead of the current asymptotic convergence results. Theorem B.4 and Theorem B.6. include the non-asymptotic convergence rates. It is natural to supplement the non-asymptotic linear convergence rates under the strongly convex condition.

**Strengths And Weaknesses:**

This submission is written with clear language and structure. It has the following strengths:

1. The authors proposed a multi-fidelity line-search scheme for zeroth order optimization methods. The authors also proved a sub-linear convergence rate of $1/k$ under standard assumptions, which is the same convergence order of the classical gradient descent method with standard backtracking line search scheme.

2. The authors developed the BF-SSD algorithm, a stochastic zeroth-order optimization method with a bi-fidelity line search that allows for choosing the approximation quality of the gradient by tuning $l$. The authors proved the convergence results of SSD with line search under three separate scenarios: strongly convex, convex, and non-convex.

3. Besides those theoretical convergence rates, the authors also compared the performance of BF-SSD with other zeroth-order optimization methods on one synthetic function and the three real-world applications objective function. The empirical results from numerical experiments are consistent with the theoretical analysis.

This submission has the following weaknesses:

1. Assumption 2.3 is a bit strong. The authors need to explain more about this assumption 2.3. The authors need to discuss this Assumption 2.3 in the examples of section 2.4.

2. The convergence results of SSD with line search under three separate scenarios: strongly convex, convex, and non-convex should be moved from Appendix B to the main part of the submission as important theoretical contributions.

3. The authors should also add the Nesterov accelerated gradient descent method in the numerical experiments.

4. The authors should change the presentation of Theorem B.2. to add the linear convergence rates.

---

> ### Author Response · Authors · 2025-08-15
>
> > The authors need to explain more details about Assumption 2.3. In section 2.4, the authors should explain more about how those concrete function examples satisfy the Assumption 2.3.
>
> First, we note a possible confusion with the old wording of Assumption 2.3, as another reviewer pointed out as well. Specifically, it is not really an assumption but a parameter setting. We have revised it to be more clear.
>
> Second, we have updated the concrete examples in section 2.4 (the quadratic objective and full vs minibatch objective) to be more explicit about how they fit the assumptions.
>
> > The authors need to move the convergence rates of SSD with line search from Appendix B to the main part of the submission. The authors also need to change the order: first introduce the results of non-convexity, then the results of convexity and finally the results of strong convexity.
>
> Our motivation for putting the (non-bi-fidelity) SSD convergence rate in the appendix is so that we don’t distract from our main message about bi-fidelity, and because other than the line search, this has already been studied in previous papers. We intentionally chose the streamlined narrative to help readers.   But we understand the reviewer’s point as well, so we have put a representative result from the appendix into the main paper.
>
> Similarly, we preferred to do the ordering of (strong convexity) then (convexity) then (non-convexity) because this is ordered by the strength of the results. We feel this is a personal preference. If all the reviewers strongly disagree with our organization and make a good case, we are happy to adjust..
>
> > Please supplement the performance of accelerated gradient descent and compare it with all other zeroth-order optimization methods in the numerical experiments.
>
> Following your recommendation, we have incorporated Nesterov gradient descent with a decaying learning rate as an additional baseline and compared it against all other methods in our numerical experiments. The updated results show that the accelerated method is indeed a competitive baseline, particularly in the first two experiments where the problem complexity is lower. However, as the complexity of the optimization landscape increases in our subsequent experiments, the advantages of our SSD approach become much more pronounced. This comparison effectively highlights SSD's robustness in navigating more challenging and complex problems. We have updated all relevant figures and discussion in the experimental section to reflect this new comparison.
>
> > The authors should add the linear convergence rates in the strongly convexity condition in Theorem B.2. instead of the current asymptotic convergence results. Theorem B.4 and Theorem B.6. include the non-asymptotic convergence rates. It is natural to supplement the non-asymptotic linear convergence rates under the strongly convex condition.
>
> Good suggestion: we have done so now.

---

### Review · Reviewer_iGJk · 2025-08-05

**Summary Of Contributions:**

This paper proposes novel methods for black box optimization in the bi-fidelity framework, where the function to be optimized can be accessed through high-fidelity but expensive and low-fidelity but cheap oracles.
These new methods are based on line-search, which aim to find the best possible step size using backtracking, on a surrogate loss constructed using low-fidelity oracles together with a linear approximation of the function based on a few high-fidelity oracles.
A convergence result is given in the case where the gradient estimate is the gradient itself, using an Armijo condition.
An additional method based on stochastic subspace descent (SSD) is then provided and studied empirically.
Examples of low-fidelity oracles are provided, together with a numerical study on a synthetic function, on kernel ridge regression, adversarial attacks in machine learning, and soft-prompting for language models.

**Audience:**

Yes

**Broader Impact Concerns:**

No broader impact concerns.

**Claims And Evidence:**

Yes

**Requested Changes:**

The paper already presents nice ideas, and the empirical evaluation is already quite complete and relevant to the discussion.
Most of my concerns lie in the theoretical part, with some results that may be incorrect:
1. Assumption 2.3 seems to assume that Equation (2.4) holds, but Equation (2.4) is itself a result of a Lemma that assumes Assumption 2.3. In this sense, what does this assumption mean? Could the author clarify Assumption 2.3 and make it self-contained without reference to the result from a further lemma?
2. Remark 2.5 looks erroneous, as it seems that it relies on Assumption 2.3, together with the assumption that $\|\|v_k\|\| = \Omega(1)$, which is not true as $k \rightarrow \infty$. Could the authors provide a proof of this remark and clarify what this result means? More specifically, it seems there is kind of a "homogeneity issue", since $WL^2$ is a product of a Lipschitz and two smoothness constants, and $L$ is a smoothness constant, making the sum $WL^2 + L$ look disturbing.
3. In Remark 2.5, it is claimed that the method allows for reducing the number of high-fidelity oracle calls, but the second term $DL/\epsilon$ seems to be the complexity of vanilla black-box GD (with good estimators of the gradients). How is that an improvement over this algorithm in terms of oracle calls?
4. In numerical experiments, all results are presented as "equivalent number of HF oracle calls". While this metric makes sense, would the same observations still hold if changing the x-axis, e.g., for wall-clock time? One could expect that the proposed method can increase the time of the training itself.
5. In practice, how does one choose $n_k$?

**Strengths And Weaknesses:**

**Strengths**
1. The proposed framework allows to efficiently use low-fidelity oracles in a clever way: it is used to construct an approximate surrogate loss, which is in turn used to tune the step-size using backtracking line search.
2. A theoretical convergence result is given, showing faster convergence of the proposed method.
3. The backtracing line search method is then applied to the stochastic subspace descent algorithm, showing promising results.
4. An important effort has been put into (i) describing what could be used as low-fidelity oracles, and (ii) studying the efficiency of the method on a wide range of problems. Numerical experiments are quite strong and show that the proposed method is quite promising for more practical use.

**Weaknesses**
1. The study of the number of oracles calls presented in Remark 2.5 may be incorrect: it seems to rely on the bound on $n_k$ presented in Assumption 2.3, which is $n_k \ge \frac{WL(1+c)\alpha_{max}}{c\beta\|\|v_l\|\|^2}$, claimed to be $n_k = O(WL)$, which is not necessarily true as $v_k$ goes to zero (which is expected as it is typically the gradient).
2. The claim in Remark 2.5 does not seem to say that the number of oracle calls can be reduced, since the second term is already the number of oracle calls of black-box GD.
3. Theoretical results could be stated with more rigor: the assumptions that are used in Theorem 2.4 should be explicitly mentioned in the theorem statement, and Remark 2.5 should be given with a proper proof. In particular, the theorems only hold when $v_k$ is chosen to be the gradient, which is quite limiting for studying a black-box method.
4. It is not very clear from the current result how the precision of the low-fidelity oracle affects the results: one could expect to see that if a very good low-fidelity oracle exists, there is a speed-up that depends on the ratio between the complexity of the high-fidelity and the low-fidelity oracles.

---

> ### Author Response · Authors · 2025-08-15
>
> > Assumption 2.3 seems ... lemma?
>
> We agree this looks circular, and we have updated the paper to clarify. We were trying to convey that Assumption 2.3 is only a sufficient condition for Eq (2.4) to hold, and not a necessary condition, but the phrasing was confusing. We think the new wording is now much more clear and thank the reviewer for pointing this out.
>
> > Remark 2.5 ... look disturbing.
>
> Regarding the first issue, we are not looking at $k\to\infty$ but rather the smallest k until the norm of the gradient squared is less than epsilon. So until then, the norm of $\lVert v_k\rVert$ is not small, so this actually is similar to $\lVert v_k\rVert=\Omega(1)$, but it is justified.
>
> Regarding the homogeneity issue, we agree that this is unexpected (e.g., that if we had units, this would seem to contradict dimensional analysis). However, it’s not incorrect and is simply a byproduct of our assumptions.  It’s always awkward to quantify how good a low-fidelity model is, and we used one method (Assumption 2.2) because it is reasonable and gives straightforward results, but we would be happy to use a better method to quantify the low fidelity if the reviewer has one in mind.
>
> > In Remark 2.5, it is ... in terms of oracle calls?
>
> Good catch, you’re right that it is not an improvement. We said “reduced” but didn’t specify with respect to what it is reduced, and you are correct that it doesn’t beat vanilla 0th order GD. We have removed that confusing sentence entirely.
>
> > In numerical experiments, all results are presented as "equivalent number of HF oracle calls". While this metric makes sense, would the same observations still hold if changing the x-axis, e.g., for wall-clock time? One could expect that the proposed method can increase the time of the training itself.
>
> Our primary use of "equivalent number of high-fidelity (HF) oracle calls" as the cost metric is rooted in its widespread adoption as a standard in the black-box optimization and multi-fidelity modeling literature. This convention provides a clear, implementation-independent basis for comparison. For example, the original Stochastic Subspace Descent (SSD) paper [1] uses the number of function calls as the x-axis in its core results (Figures 2, 3, and 6). Similarly, Ng and Willcox [2] define cost as "computational effort," which they specify is the "equivalent number of high-fidelity model evaluations" (see discussion above Eq. (5)). Notice that, for the practical cases where the evaluation of the HF function is computationally expensive, the overall wall-clock time is dominated by the HF function calls and the rest of the optimization incurs negligible cost.
>
> To directly address your question, we conducted a new set of experiments measuring performance against wall-clock time. The results, available in the link https://limewire.com/d/cgXQn#LrFI7AyFC4, demonstrate a strong alignment with the results presented in the paper. The relative performance of all methods remains consistent whether the x-axis represents equivalent HF calls or wall-clock time. This confirms that the computational overhead of our method does not alter the paper's conclusions.
>
> We have updated the supplementary files to include the code used for these wall-time calculations (see `util/OPT_utilities.py`, lines 22 and 37). To maintain consistency with the established literature and ensure the main paper remains focused, we will retain the figures using HF calls. However, based on your valuable feedback, we have added a comment to the experiments section (above section 4.1) clarifying that the trends shown using equivalent HF calls are directly representative of the performance measured by wall-clock time.
>
> [1] David Kozak, Stephen Becker, Alireza Doostan, and Luis Tenorio. A stochastic subspace approach to gradient-free optimization in high dimensions. Computational Optimization and Applications, 79(2):339–368, 2021.
>
> [2] Leo W. T. Ng and Karen E. Willcox. Multifidelity approaches for optimization under uncertainty. International Journal for numerical methods in Engineering, 100(10):746–772, 2014.
>
> > In practice, how does one choose $n_k$?
>
> Parameter selection is often an issue for any algorithm. In practice, we kept simple ($n_k=1$, i.e., one additional function evaluation in addition to the evaluation at the incumbent point) as this already worked well.  We didn’t want to explore the issue too much at such a preliminary stage, but we have some basic thoughts. In particular, taking inspiration from trust-region methods, we might start with a small $n_k$, build the surrogate, and predict the decrease, then compare to the actual function value decrease. If the predicted decrease is significantly over-optimistic, then we go back and rebuild the surrogate with a larger $n_k$.  We’ve added a few thoughts along these lines into the main text at the end of section 3, right before the empirical experiments section.

---

### Decision · Action_Editor_BDQy · 2025-09-18

**Recommendation:** Accept with minor revision

**Additional Comments:**

Authors should take into account these comments for the camera ready version:

- Theoretical analysis is a bit weaker than numerical analysis and relies on strong assumptions that may be atypical in zeroth-order optimization. In particular, the way Assumption 2.3 is stated is unusual, as it refers to the iterations of the algorithm: it seems that it is rather a way of setting the algorithm's hyperparameter $\eta_k$ rather than a proper assumption.
- Convergence results are given under the strong assumption that the gradient estimate is the actual gradient, which is strong for the considered setting, and should be mentioned explicitly in the statement of Theorem 2.4.
- Remark 2.5 should be corrected: as per the authors' claim, it holds as long as  $\|v_k\|^2 \geq \epsilon$ (which is the case during the optimization), which authors claim to be $1 / \|v_k\|^2 = O(1)$, but is rather $1 / \|v_k\|^2 = O(1/\epsilon)$: this has an impact on  Equation 2.2, where I believe the first term of the right-hand-term should be in $1/\epsilon^2$  rather than $1/\epsilon$.

**Audience:**

Yes

**Audience Explanation:**

There is consensus that the work will be of interest to TMLR readers.
Efficient 0th-order optimization is relevant to both machine learning and scientific computing communities.

**Claims And Evidence:**

Yes

**Claims Explanation:**

All reviewers agree that the claims of the paper are supported by empirical evidence, with experiments covering diverse probles.
Theoretical analysis is less convincing: several reviewers highlight that convergence guarantees rely on strong and somewhat atypical assumptions. Nevertheless, the authors have addressed most points in revisions.

---

> ### Author Response · Authors · 2025-09-28
>
> We sincerely thank the comments and excited about the final decision. We have uploaded the camera-ready version to address raised concerns.
>
> * **Regarding Assumption 2.3:** We agree and rephrased the assumption.
>
> * **Regarding Theorem 2.4:** We have now explicitly stated the condition that the gradient estimate is the true gradient within the statement of Theorem 2.4.
>
> * **Regarding Remark 2.5 and Equation 2.2:** We have corrected Remark 2.5 to reflect that $1/\|v_k\|^2 = O(1/\epsilon)$. Consequently, we have updated Equation 2.2, and the first term on the right-hand side is now correctly stated to be in $O(1/\epsilon^2)$.

---

> > ### Comment · Action_Editor_BDQy · 2025-09-29
> >
> > Dear authors,
> >
> > I believe you forgot to de-anonymize the camera ready version
> >
> > AE